# MAGNITUDER LAYERS FOR IMPLICIT NEURAL REPRESENTATIONS IN 3D

## ABSTRACT

Improving the efficiency and performance of implicit neural representations in 3D, particularly Neural Radiance Fields (NeRF) and Signed Distance Fields (SDF) is crucial for enabling their use in real-time applications. These models, while capable of generating photo-realistic novel views and detailed 3D reconstructions, often suffer from high computational costs and slow inference times. To address this, we introduce a novel neural network layer called the "magnituder", designed to reduce the number of training parameters in these models without sacrificing their expressive power. By integrating magnituders into standard feed-forward layer stacks, we achieve improved inference speed and adaptability. Furthermore, our approach enables a zero-shot performance boost in trained implicit neural representation models through layer-wise knowledge transfer without backpropagation, leading to more efficient scene reconstruction in dynamic environments.

## 1 INTRODUCTION

In recent years, the integration of advanced machine learning techniques into robotics has transformed the field of robotic manipulation and trajectory planning. As robots increasingly operate in unstructured environments, the ability to perceive and interact with complex scenes becomes essential. Traditional robotic perception methods often rely on explicit geometric models, which can limit flexibility and adaptability. In contrast, implicit neural representations such as Neural Radiance Fields (NeRF) (Mildenhall et al., 2020) and incremental Signed Distance Fields (iSDF) (Ortiz et al., 2022) have emerged as powerful tools that offer robust ways to model 3D scenes with high fidelity.

NeRF, initially developed for novel view synthesis, uses multiple feed-forward layers (FFLs) to encode 3D scenes into a continuous volumetric representation, enabling photo-realistic rendering from arbitrary viewpoints. Its ability to learn intricate scene details from sparse data makes NeRF particularly appealing for robotics applications, especially in dynamic or poorly structured environments. On the other hand, iSDF provides a versatile framework for capturing the geometric properties of objects, facilitating real-time collision detection and path planning. By representing surfaces as implicit functions, iSDF seamlessly integrates shape information into robotic control systems.

Despite the exciting opportunities that NeRF and iSDF offer for enhancing robotic manipulation and trajectory planning, their practical deployment in real-world scenarios still requires improvements in training and inference speed. While iSDF provides real-time mapping, it relies on known camera poses, which adds computational overhead in real-world applications. Accelerating both training and inference can make the entire system faster, expanding the potential for real-time applications.

In real-world applications, robots often encounter novel objects and unpredictable conditions. Rapid adaptation to new scenarios is crucial for maintaining operational efficiency and safety. Quickly retraining models or adapting them on-the-fly can significantly improve a robot's performance in tasks such as grasping or navigating cluttered spaces. Slow training cycles may result in outdated models that fail to accurately represent the current environment, reducing reliability and effectiveness.

Moreover, inference speed is vital for real-time decision-making in robotic systems. Delays in processing visual and spatial data can negatively impact a robot's ability to perform timely actions, which is particularly important in fast-paced environments. For instance, in collaborative settings where robots work alongside humans, latency can create safety risks and hinder task efficiency.

Thus, optimizing inference times while maintaining high accuracy is essential for deploying these advanced representations in practical applications.

The main computational block for NeRF and iSDF are FFLs. To reduce the computational complexity, various methods have been proposed that reduce the burden of FFLs by exploiting explicit representations (Chen et al., 2022; Sun et al., 2022; Müller et al., 2022; Yu et al., 2021; Fridovich-Keil et al., 2022; Liu et al., 2020; Barron et al., 2023), reducing the number of network inferences through sample pruning with additional occupancy flags (Sun et al., 2022; Müller et al., 2022; Li et al., 2023a), or introducing a novel volume rendering procedure (Han et al., 2024).

In this work, we address this challenge by introducing a novel neural network layer, called *magnituder* that can be combined with standard FFLs in an implicit neural representation. The magnituders disentangle the processing of weights and inputs to the layer, only connecting them at the end through simple linear operations. The magnituders use structured transformations of the input to the layer that depend solely on the magnitude of the input (hence their name). This design allows for a reduction in the training parameters of the FFL blocks while maintaining the same level of expressivity as a standard FFL.

To summarize, our contributions in this work are the following :

- We introduce a novel random feature mechanism called *magnituders* that optimizes the computation of FFLs in implicit neural representations (Section 3).
- We demonstrate how this mechanism can approximate FFL with commonly used activation functions like ReLU and Softplus (Section 4.1).
- We showcase the versatility of our method across a wide range of scenes and a variety of implicit neural representations (Section 4.2).
- We distill existing MLP layers in an implicit neural representation with our magnituders which are trained analytically *without backpropagation*. Our method allows for zero-shot integration of our novel layer with pretrained models, improving inference speed without costly retraining. To the best of our knowledge, this is the first successful application of knowledge distillation without backpropagation for implicit neural representations, demonstrating its effectiveness in downstream tasks (Section 4.3).

Note that our method is orthogonal to the sparse sampling and the partitioning techniques and can be combined in such to further reduce the computational footprint.

## 2 RELATED WORK

Neural Radiance Fields (NeRF) (Mildenhall et al., 2020) have transformed the synthesis of novel views from posed 2D images, offering impressive quality in rendering complex 3D scenes. However, both training and rendering processes remain time-consuming, prompting extensive research aimed at accelerating these tasks, by integrating explicit representations (Liu et al., 2020; Yu et al., 2021; Hu et al., 2022; Tancik et al., 2022; Takikawa et al., 2021; Chen et al., 2022; Sun et al., 2022; Müller et al., 2022; Fridovich-Keil et al., 2022; Xu et al., 2023; Barron et al., 2023; Takikawa et al., 2023; Hu et al., 2023), dividing a scene into smaller blocks (Reiser et al., 2021; Tancik et al., 2022; Turki et al., 2022; Xiangli et al., 2022; Turki et al., 2023), caching (baking) the implicit functions (Garbin et al., 2021; Hedman et al., 2021; Chen et al., 2023; Reiser et al., 2023), and devising some novel tweaks (Lindell et al., 2021; Han et al., 2024). Given their effectiveness, there has also been significant work extending NeRF to accommodate dynamic (Pumarola et al., 2021; Park et al., 2021a;b; Li et al., 2022; Weng et al., 2022; Fang et al., 2022; Park et al., 2023; Cao & Johnson, 2023; Fridovich-Keil et al., 2023; Li et al., 2023c; Wang et al., 2023d; Shao et al., 2023; Attal et al., 2023; Wang et al., 2023a; Lin et al., 2023; Xu et al., 2024), unbounded (Wang et al., 2021b; Barron et al., 2022; Choi et al., 2023; Barron et al., 2023; Wang et al., 2023c), and challenging appearances scenarios (Martin-Brualla et al., 2021; Ichnowski et al., 2021; Mildenhall et al., 2022; Bemana et al., 2022; Verbin et al., 2022; Guo et al., 2022; Fujitomi et al., 2022; Zhan et al., 2023; Warburg et al., 2023; Lee et al., 2023; Kim et al., 2023; Goli et al., 2024; Verbin et al., 2024; Ren et al., 2024).

Signed Distance Fields (SDF) are scalar fields that represent the distance to the nearest surface point, crucial in robotics for tasks such as environment mapping, collision avoidance, and trajectory opti-

mization (Zucker et al., 2013; Toussaint, 2009; Schulman et al., 2014; Mukadam et al., 2018; Dong et al., 2018; 2016; Chaplot et al., 2021; Das & Yip, 2020; Verghese et al., 2022; Zhang et al., 2023; Johari et al., 2023; Deng et al., 2023; Yan et al., 2023; Zhu et al., 2024). In computer vision, Truncated Signed Distance Fields (TSDF) are employed in SLAM systems, as traditional SDF computations can be prohibitively expensive in real time (Newcombe et al., 2011; 2015). Several approaches have emerged to facilitate real-time SDF construction due to its importance in robotics (Ortiz et al., 2022; Han et al., 2019; Oleynikova et al., 2017), while others have addressed challenges in constructing SDF for complex environments (Finean et al., 2021; Reijgwart et al., 2020; Geng et al., 2023; Qiu et al., 2024). To get a photo-realistic quality of scene reconstruction, recent works replace the volume density in NeRF formulations with SDF, using well-designed SDF-to-volume density conversion formulas (Yariv et al., 2020; Wang et al., 2021a; Yariv et al., 2021). Further acceleration is achieved by combining explicit grid structures with implicit SDF (Yu et al., 2022; Wang et al., 2023e; Li et al., 2023b; Wang et al., 2023b; Rosu & Behnke, 2023), populating thin shells near the zero level set (Wang et al., 2023f), or baking SDF into a high-quality triangle mesh (Yariv et al., 2023).

A common challenge in training neural networks is that their space and time complexity scale quadratically with the size of their hidden layers. This challenge is also inherited by implicit neural representations. One of the ways to tackle this issue is via dimensionality reduction à la Johnson-Lindenstrauss Transform (or JLT) (Dasgupta & Gupta, 2003; Dasgupta et al., 2010; Ailon & Liberty, 2013) or kernel methods (Rahimi & Recht, 2007). Kernel methods have a rich history, spanning the linearization of 2-layer neural networks (Cho & Saul, 2009; 2011), Neural Tangent Kernels (NTK)(Jacot et al., 2018), and the linearization of attention mechanisms in Transformers (Choromanski et al., 2020). Kernel methods using random feature mechanisms have shown promise in reducing the computational complexity of neural networks (Choromanski et al., 2020; Rahimi & Recht, 2007). However, these approaches are not directly applicable to the MLP layers that are the backbone of NeRF and iSDF systems. To address the computational complexity of FFLs, Sehanobish et al. (2024) introduced the Universal Random Feature (URF), which disentangles weights and inputs in specific MLP layers. Their method relies on Fourier transforms of activation functions, limiting its applicability to commonly used activation functions in machine learning.

## 3 MAGNITUDERS

In this section we define our magnituder-layers (MAG-layers in short). Consider a layer $\mathcal{M} : \mathbb{R}^d \to \mathbb{R}^l$ taking as input a $d$-dimensional vector, outputting an $l$-dimensional vector and given as follows:

$$\mathcal{M}_{\mathbf{W},\theta}(\mathbf{x}) = \mathbb{E}\left[\mathbf{W}\mathbf{v}_\theta(\|\mathbf{x}\|_2)\right], \tag{1}$$

where $\mathbf{W} \in \mathbb{R}^{l \times d}$ is a learnable matrix and $\mathbf{v}_\theta : \mathbb{R} \to \mathbb{R}^d$ is a map operating on the magnitude (length) of the input $\mathbf{x}$. This transformation is potentially learnable (with a set of learnable parameters $\theta$) or probabilistic (in the latter setting, $\theta$ encode the parameters of the distribution $\mathcal{D}_\theta$ used to sample the values of $\mathbf{v}_\theta(\mathbf{x})$). The expectation is with respect to $\mathcal{D}_\theta$. In practice, in the probabilistic setting, $\mathcal{M}_{\mathbf{W},\theta}(\mathbf{x})$ is not computed exactly, but unbiasedly approximated by sampling.

Magnituders disentangle the processing of the input $\mathbf{x}$ and weights $\mathbf{W}$, only to connect them via simple linear matrix-vector multiplication at the end. In magnituders, randomness is introduced via multiplications of $\mathbf{x}$ with a Gaussian matrix $\mathbf{G} \in \mathbb{R}^{m \times d}$ where the entries are drawn iid from $\mathcal{N}(0, 1)$, for some $m > 0$ which are subsequently followed by purely deterministic transformations. The parameter $m$ is called the number of random features in a MAG-layer. Under this randomization strategy, taking a kernel perspective, magnituders can be interpreted as outputting a vector of approximate kernel values $\mathrm{K}(\mathbf{w}_i, \mathbf{x})$ for $i = 1, ..., l$, where kernel $\mathrm{K} : \mathbb{R}^d \times \mathbb{R}^d \to \mathbb{R}$ is defined as:

$$\mathrm{K}_{\mathrm{MAG}}(\mathbf{u}, \mathbf{v}) \stackrel{\text{def}}{=} \sum_{i=1}^d u_i \mathbb{E}[f_{\theta_i}(\|\mathbf{v}\|_2 g)], \tag{2}$$

where $g \stackrel{\text{iid}}{\sim} \mathcal{N}(0, 1)$ and functions $f_{\theta_1}, ..., f_{\theta_d} : \mathbb{R} \to \mathbb{R}$ are deterministic. This comes directly from the fact that for $\mathbf{g} \sim \mathcal{N}(0, \mathbf{I}_d)$, we have: $\mathbf{v}^\top \mathbf{g} \sim \|\mathbf{v}\|_2 \times \mathcal{N}(0, 1)$.

The above kernel can be efficiently and accurately approximated by *quasi Monte Carlo* methods (QMCs).

### 3.1 MAGNITUDERS AND QMCS

A straightforward approach to estimating kernels from Equation 2 is through Monte Carlo methods, which is effectively implemented in versions of the magnituder layers that leverage Gaussian matrices $\mathbf{G}$.

To be more specific, the kernel from Eq. 2 is approximated as follows : For $\mathbf{g}_1, ..., \mathbf{g}_m \overset{\text{iid}}{\sim} \mathcal{N}(0, \mathbf{I}_d)$:

$$\mathrm{K}_{\mathrm{MAG}}(\mathbf{u}, \mathbf{v}) \approx \sum_{i=1}^{d} u_i f_{\theta_i}(\mathbf{v}^\top \mathbf{g}_i), \tag{3}$$

A standard technique to reduce the variance for approximators based on ensembles of Gaussian vectors is via the so-called *orthogonal random features* or ORFs (Yu et al., 2016). The ensembles of iid vectors are replaced by block-orthogonal ensembles, where within each $d$-element block vectors have marginal Gaussian distributions $\mathcal{N}(0, \mathbf{I}_d)$, yet different vectors in each block are exactly orthogonal. Blocks are constructed independently. ORFs are popular since they can be easily constructed. For magnituders we have the following variance reduction lemma:

**Lemma 3.1.** *Denote by* $\widehat{\mathrm{K}}_{\mathrm{MAG}}^{\mathrm{iid}}(\mathbf{u}, \mathbf{v})$ *and estimator from the RHS of Eq. 3 appying iid ensemble:* $\mathbf{g}_1^{\mathrm{iid}}, ..., \mathbf{g}_m^{\mathrm{iid}}$ *and by* $\widehat{\mathrm{K}}_{\mathrm{MAG}}^{\mathrm{ort}}(\mathbf{u}, \mathbf{v})$ *its variant leveraging block-orthogonal ensemble (ORFs):* $\mathbf{g}_1^{\mathrm{ort}}, ..., \mathbf{g}_m^{\mathrm{ort}}$. *If functions* $f_{\theta_i}$ *are analytic with non-negative Taylor coefficients and* $u_i \geq 0$ *for* $i = 1, ..., m$ *then:*

$$\mathrm{Var}(\widehat{\mathrm{K}}_{\mathrm{MAG}}^{\mathrm{ort}}(\mathbf{u}, \mathbf{v})) \leq \mathrm{Var}(\widehat{\mathrm{K}}_{\mathrm{MAG}}^{\mathrm{iid}}(\mathbf{u}, \mathbf{v})) \tag{4}$$

*Proof.* The proof follows directly from Lemma 4 and Theorem 6 in Choromanski et al. (2020). □

We observe that, empirically, even in cases where the conditions of Lemma 3.1 are not satisfied, the ORFs still provide quality gains (see Figure 19).

Finally, we would like to posit our magnituder layers in the broader context of linearizing neural networks via kernel methods. If $f_{\theta_i}$'s are deterministic functions, the right hand side of Equation 2 represents a 2-layer NN where the input layer is fixed and only $u_i$ is trainable. The linearization of such 2-layer NN is widely considered under various conditions (Cho & Saul, 2009; 2011; Ghorbani et al., 2020; Neal, 1996). The notable difference between the previous works and our work is that $\mathrm{K}_{\mathrm{MAG}}$ cannot be modeled as a standard neural network i.e. matrix-vector multiplications followed by point-wise nonlinear transformations, so such techniques can not be readily applied here.

### 3.2 CONNECTIONS WITH SNNK

We now discuss our magnituder layer with another similar mechanism SNNK introduced by Sehanobish et al. (2024). Like magnituders, SNNKs also disentangle the weights and the inputs by leveraging the so-called Universal Random Features (URFs). However there are some key differences, which we detail below:

The SNNK-layers are defined as follows:

$$\mathrm{SNNK}_{\mathbf{W}}(\mathbf{x}) = \mathbb{E}[\Phi(\mathbf{W})\Psi(\mathbf{x})], \tag{5}$$

where: (1) probabilistic map $\Phi : \mathbb{R}^{l \times d}$ is obtained by applying probabilistic maps $\Phi^i : \mathbb{R}^d \to \mathbb{R}^d$ for $i = 1, ..., l$, independently on the verticalized rows $\mathbf{w}_1, ..., \mathbf{w}_l$ of $\mathbf{W}$ respectively and furthermore, (2) $\Psi : \mathbb{R}^d \to \mathbb{R}^d$ is another probabilistic map.

Thus the randomness is introduced in both the weight and the input tower. And the kernel that is associated to SNNK can be defined as :

$$\mathrm{K}_{\mathrm{SNNK}}(\mathbf{u}, \mathbf{v}) \overset{\text{def}}{=} \mathbb{E}_{\mathbf{g} \in \mathcal{N}(0, \mathbf{I}_d)}[\Phi(\mathbf{u}^\top \mathbf{g})\Psi(\mathbf{v}^\top \mathbf{g})] \tag{6}$$

which can be estimated using QMC methods via the following approximation :

$$\mathrm{K}_{\mathrm{SNNK}}(\mathbf{u}, \mathbf{v}) \approx \frac{1}{m} \sum_{i=1}^{m} \Phi(\mathbf{u}^\top \mathbf{g}_i)\Psi(\mathbf{v}^\top \mathbf{g}_i) \tag{7}$$

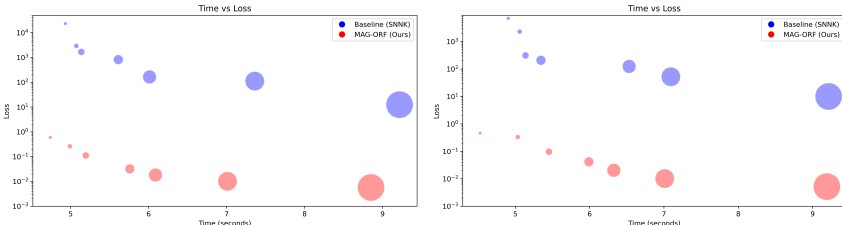

Figure 1: Approximation capability of our MAG layer. Here we compare with the SNNK layer as our baseline. The size of the dots represents the number of training parameters and for fairness, we made sure that the MAG layer and the SNNK has the same number of training parameters. (Left) : Approximating a ReLU-Linear layer. (Right) : Approximating a Softplus-Linear layer.

However, rigorous theoretical results regarding variance reduction from ORFs for the estimators defined in the RHS of Eq. 7 are known only for very specific nontrivial functions $\Phi$ and $\Psi$, i.e. the exponential map: $z \to \exp(c \times z)$ (Choromanski et al., 2020) and sgn-function $z \to \text{sgn}(z)$ (Choromanski et al., 2017). Moreover, we believe that the explicit dependence on $\|\mathbf{x}\|_2$ in magnituders, enables them to outperform SNNKs in applications leveraging implicit neural-network-based representations, as we show in Sec. 4.1 (Figure 1).

## 4 EXPERIMENTS

In this section, we present a comprehensive empirical evaluation of our MAG-layers. Specifically, we design experiments to answer the following research questions:

- (Q1) How accurate are MAGs in approximating some commonly used FFLs?
- (Q2) How can MAGs be used to speed up various existing INRs?
- (Q3) How can MAGs be used to improve inference speeds of an already trained INR?

In all our experiments, we use the MAG-layer of the form $\mathbf{M_W}(\mathbf{x}) = \mathbf{W}f(\mathbf{Gx})$, where $\mathbf{W} \in \mathbb{R}^{d' \times m}$ is a trainable matrix, $\mathbf{G} \in \mathbb{R}^{m \times d}$ is a random matrix, $d$ (resp. $d'$) is the dimension of the input (resp. output), $m$ is the number of random features, and $f = \text{ReLU}$ applied point-wise. By choosing $m << d$, we can achieve a reduction in training parameters while maintaining comparable performance, as we will demonstrate in these experiments. Additional experiments including detailed ablation studies can be found in Appendix D.

### 4.1 SYNTHETIC EXPERIMENTS

First, we justify our choice of $f$ by demonstrating the superiority of our magnituder (MAG) layers in approximating both ReLU-linear and SoftPlus-linear layers. These activations are selected because they are commonly used in downstream applications. Additionally, we show the performance boost over the existing random feature mechanism (SNNK) (Sehanobish et al., 2024). We replicate this experiment over 10 random seeds and present the average of these runs in Figure 1. More details can be found in Appendix C.1. Furthermore, we support our choice of using ORFs (i.e. $\mathbf{G}$ is an orthogonal random matrix) with the results presented in Figure 19.

Thus in the subsequent sections, we will only use ORFs in our MAG-layers.

### 4.2 MAG-LAYERS FOR IMPLICIT NEURAL REPRESENTATIONS

In this section, we show how MAG layers can be seamlessly injected into various implicit neural representation methods including NeRF (Mildenhall et al., 2020), Zip-NeRF Barron et al. (2023), D-NeRF (Pumarola et al., 2021) and iSDF (Ortiz et al., 2022), to accelerate existing methods. Additionally, we introduce novel design choices for these models to further enhance inference speeds.

Recent works have shown training the projection matrix can lead to accuracy gains (Chowdhury et al., 2022; Zhang et al., 2024). We call those variants MAG-layers with trainable $\mathbf{G}$.

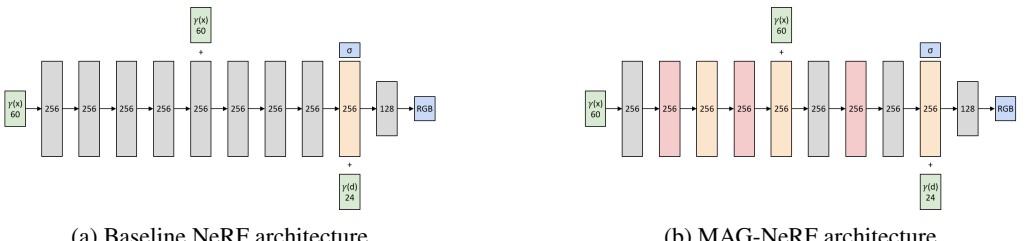

Figure 2: Results of rendering through NeRF versus MAG-NeRF: *drums* (left) and *ficus* (right). The Ground Truth column shows the reference image, followed by the rendered results.

Table 1: Comparison between NeRF and our MAG-NeRF on the Synthetic NeRF dataset (Mildenhall et al., 2020). We report the mean results for all 8 scenes. The inference time refers to the approximate time for forward passes of each model to render a single $400 \times 400$ image. Our MAG-NERF is only 1% behind the baseline while being 24% faster.

| Metric | PSNR ($\uparrow$) | SSIM ($\uparrow$) | LPIPS ($\downarrow$) | Inference time ($\downarrow$) (s) | Model Size ($\downarrow$) (MB) |
|---|---|---|---|---|---|
| NeRF | 30.45 | 0.950 | 0.032 | 2.36 | 3.27 |
| MAG-NeRF (ours) | 30.10 | 0.945 | 0.035 | 1.79 | 2.27 |

(a) Baseline NeRF architecture        (b) MAG-NeRF architecture

Figure 3: Network architecture of the baseline NeRF and our MAG-NeRF model. **Black** denotes Linear ReLU layer, **Orange** is Linear without activation, **Red** is the MAG layer, + is concatenation, and the activation for the RGB output is Softmax. Note that MAG layer followed by the Linear layer can be compressed for more efficient network inference.

### 4.2.1 NERF EXPERIMENTS

In this section, we focus on integrating MAG-layers into NeRF models (Mildenhall et al., 2020). While NeRF has demonstrated impressive capabilities in generating photo-realistic scenes, it suffers from long training and inference times. Thus, to accelerate inference times and reduce model sizes, we replace three MLP layers in the baseline NeRF architecture with our random feature mechanism, which we refer to as MAG-NeRF (see Figure 3). Our design choice enables the "bundling process", allowing subsets of layers to be grouped together to create a lean, efficient model without sacrificing accuracy (see Appendix A).

We evaluate both baseline NeRF and MAG-NeRF using the PyTorch implementation of NeRF (Yen-Chen, 2020) on the Synthetic NeRF dataset (Mildenhall et al., 2020). We train MAG-NeRF in the same manner as the baseline NeRF by running 200k iterations and rendering the test images at half resolution. During inference, we apply a bundling process to the MAG-NeRF network by collapsing the MAG layer with any subsequent Linear layers before nonlinear activation. When concatenation is involved, we decompose the weight matrix into two parts and bundle them together. For instance, in Figure 3 (b), we bundle three consecutive layers ($4^{th}$-$6^{th}$ arrows: Red, Orange, and Black) by decomposing the Linear-ReLU layer (black arrow) into two parts: $60 \times 256$ and $256 \times 256$.

In Table 1 and Figure 2, we present the quantitative and qualitative results of our MAG-NeRF. We report the rendering quality metrics with PSNR, SSIM (Wang et al., 2004), and LPIPS (Zhang et al., 2018), along with the network inference time required to render an image. MAG-NeRF significantly reduces both inference time and model size by approximately **24%** and **30%**, respectively, with virtually no loss in reconstruction quality. Detailed results on each scene is presented in Tables 6, 7 and 8. Other rendered images are in Figures 10 and detailed ablation studies can be found in Figure 17 as well as in Table 14.

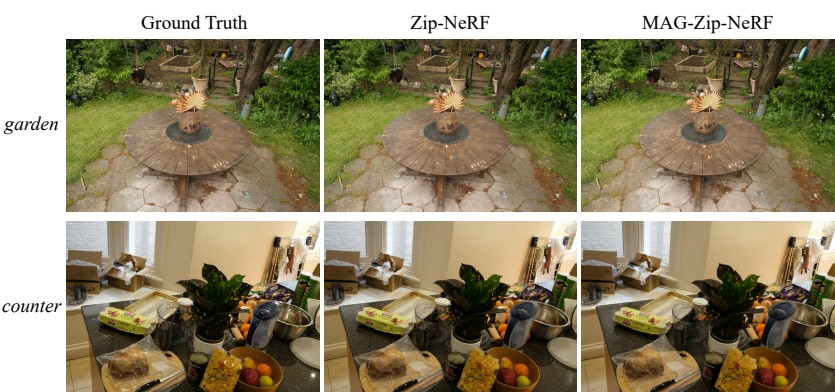

Figure 4: Rendered images from the Mip-NeRF 360 dataset (Barron et al., 2022). For each scene, the images from left to right show the Ground Truth, Zip-NeRF, and MAG-Zip-NeRF.

Table 2: Comparison between Zip-NeRF (Barron et al., 2023) and MAG-Zip-NeRF on the *360_v2* dataset (Barron et al., 2022). We report the mean results for all 7 scenes. Our trainable $\mathbf{G}$ variant is only .9% behind the baseline while being 6% faster.

| Metric | PSNR ($\uparrow$) | SSIM ($\uparrow$) | LPIPS ($\downarrow$) | Inference time ($\downarrow$) (s) |
|---|---|---|---|---|
| Zip-NeRF | 28.18 | 0.823 | 0.193 | 4.03 |
| MAG-Zip-NeRF (ours) | 27.81 | 0.815 | 0.198 | 3.78 |
| MAG-Zip-NeRF w/ trainable $\mathbf{G}$ (ours) | 27.94 | 0.818 | 0.196 | 3.78 |

### 4.2.2 ZIP-NERF EXPERIMENTS

Prior works have accelerated NeRF by integrating explicit grid structures with MLPs (Chen et al., 2022; Sun et al., 2022; Müller et al., 2022; Barron et al., 2023). By incorporating these explicit structures, networks can encode spatial information more efficiently, reducing the computational burden on MLPs. Despite these improvements, there remains room for further acceleration within the MLP architectures themselves. We address this shortcoming using our MAG layers.

We specifically focus on Zip-NeRF (Barron et al., 2023), which has demonstrated exceptional performance in unbounded scenes and relies on shallow MLPs to decode grid-based features. We use the Pytorch implementation of Zip-NeRF (Gu, 2023) for both the baseline and our modification. Similar to our approach with the baseline NeRF architecture, we identify two consecutive linear layers with ReLU activations within Zip-NeRF and replace the first of these with our MAG layer, resulting in a variant we refer to as MAG-Zip-NeRF. Since this replacement occurs in only a single place, this experiment serves as an extreme case study for the effectiveness of our method.

Figure 4 and Table 2 present the qualitative and quantitative results. Despite this minimal modification, we maintain similar rendering quality while improving the rendering speed by **6**%. Figure 11 shows the rendered images for other scenes.

### 4.2.3 DYNAMIC NERF EXPERIMENTS

For our next NeRF experiment, we focus on D-NeRF (Pumarola et al., 2021), which can synthesize novel views of dynamic scenes with complex non-rigid geometries. It represents a dynamic scene using an implicit neural network by modeling the dynamic movement, named deformation network. We choose Pytorch implementation of D-NeRF (Tang, 2022) which further integrates the hash-grid (Müller et al., 2022) to accelerate the training and rendering. We inject our MAG layer in the deformation network, thereby accelerating the rendering of dynamic NeRF (see Appendix C.4 and Figure 12 for more details).

We present the quantitative results in Table 3 with the rendered images in Figure 13. We achieve a **7**% speedup in inference while maintaining a similar reconstruction quality.

Table 3: Comparison between D-NeRF and our MAG-D-NeRF on the Dynamic NeRF dataset (Pumarola et al., 2021). As an average over 8 scenes, our trainable **G** variant is only 1% behind the baseline while being 7% faster.

| Metric | PSNR (↑) | SSIM (↑) | LPIPS (↓) | Inference time (↓) (s) |
|---|---|---|---|---|
| D-NeRF | 30.70 | 0.954 | 0.033 | 1.269 |
| MAG-D-NeRF (ours) | 30.26 | 0.948 | 0.038 | 1.177 |
| MAG-D-NeRF w/ trainable **G** (ours) | 30.32 | 0.948 | 0.038 | 1.177 |

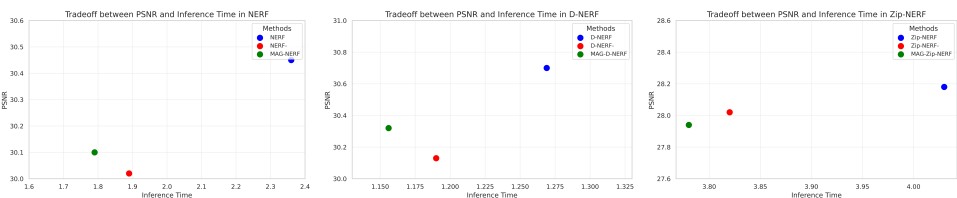

Figure 5: Comparison of MAG models with baseline models with same number of trainable parameters. In all cases, our MAG-models achieves better or similar reconstruction quality than the base models with same number of parameters while being faster. Model- refers to the baseline models with the same number of parameters as the MAG-model. Left : NERF. Middle : D-NERF. Right : Zip-NERF.

### 4.2.4 MIP-NERF 360 EXPERIMENTS

For our final NeRF experiment, we focus on Mip-NeRF 360 (Barron et al., 2022) as it is a SOTA architecture for novel view synthesis. Due to the computational expense of this experiment, we exclusively use the trainable **G** variant, as it is our most performant configuration. We observe that our model is only 2% off from the baseline while being 18% faster during inference (see Table 4).

### 4.2.5 iSDF EXPERIMENTS

In our final application, we explore iSDF (Ortiz et al., 2022), a real-time module designed to reconstruct SDF from depth sequences. This application highlights the versatility of our method, showcasing its effectiveness in real-time scenarios. The network architecture consists of a 7-layer MLP (See Fig. 14). We replace Softplus with ReLU for initial speedup and then evaluate whether further acceleration can be achieved with MAG layers (See Appendix D and Fig. 16 for justification).

We modify the original iSDF network using a similar strategy as in our NeRF experiments, resulting in our new variant, MAG-iSDF (details are provided in Appendix C.6). We evaluate the accuracy of our MAG-iSDF in ReplicaCAD (Szot et al., 2021) sequences using three metrics: average SDF error ($L1_{avg}$) over the entire scene, surface SDF error ($L1_{surf}$) on GT mesh surface, and the one-way Chamfer Distance (CD) from the reconstructed to the GT mesh. We also report the training time per step ($t_{train}$) including sample generation and forward-backward passes, and inference time ($t_{infer}$) by measuring 1000 random rays' forward pass.

In Table 5, we compare the original iSDF with MAG-iSDF. Our method is only 2% behind the baseline while being **23**% faster. We show detailed ablation results with different number of random features in Table 12. The results reveal a trade-off between computational speed and reconstruction quality (Fig. 7 (Right)). Moreover, when we visualize slices of the reconstructed SDFs on the xy-plane at $z = 70$ cm, the reconstruction quality remains quite similar (Fig. 6).

Reducing hidden dimensions is a straightforward method to accelerate training. For fair comparison, we reduce the number of hidden channels in the last layer to 32 and replace the last hidden layer with a MAG layer containing 32 random features. We evaluate this on the longest sequence, *apt_2_nav*, from the Replica dataset in iSDF. In Figure 7, we plot the inference time and Chamfer Distance. While hidden dimension reduction (DR) reduces inference time at the expense of quality, our MAG layer achieves a similar speed-up while maintaining reconstruction quality. The bundling process combines the last two layers into a single operation, leading to additional speed-up without compromising quality (Figure 7).

Table 4: Comparison between Mip-NeRF 360 (Barron et al., 2022) and MAG-Mip-NeRF 360 on the *360_v2* dataset. As an average over 7 scenes, our model is only 2% behind the baseline while being 18% faster.

| Metric | PSNR (↑) | SSIM (↑) | LPIPS (↓) | Inference time (↓) (s) |
|---|---|---|---|---|
| Mip-NeRF 360 | 27.27 | 0.733 | 0.335 | 33.15 |
| MAG-Mip-NeRF 360 (ours) | 26.81 | 0.710 | 0.363 | 27.30 |

Table 5: Quantitative results of MAG-iSDF. L1$_{avg}$ is calculated across all SDF levels, and L1$_{surf}$ is for ground truth surface points. $t_{train}$ refers to iteration time (including backward pass), and $t_{infer}$ measures forward pass inference time. Results are averaged across 6 ReplicaCAD (Szot et al., 2021) sequences with 3 random seeds. Our MAG-iSDF model achieves less than similar quality as the baseline while being 23% faster.

| Method | Trainable **G** | Distance | | | Time | |
|---|---|---|---|---|---|---|
| | | L1$_{avg}$ (↓) [cm] | L1$_{surf}$ (↓) [cm] | CD (↓) [cm] | $t_{train}$ (↓) [ms] | $t_{infer}$ (↓) [ms] |
| iSDF | NA | 8.99 | 4.08 | 2.02 | 8.24 | 1.37 |
| MAG-iSDF (ours) | No | 9.62 | 4.28 | 2.12 | 7.74 | 1.06 |
| | Yes | 9.37 | 4.17 | 2.10 | 7.81 | 1.06 |

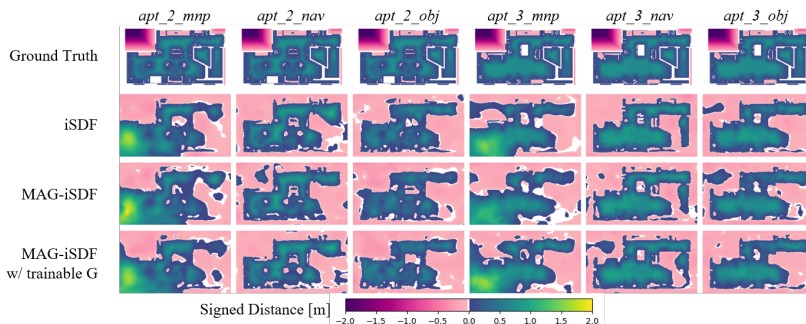

Figure 6: Visualization of reconstructed SDFs in iSDF for 6 ReplicaCAD dataset. Each figure shows the reconstructed SDF, following the colormap at the bottom.

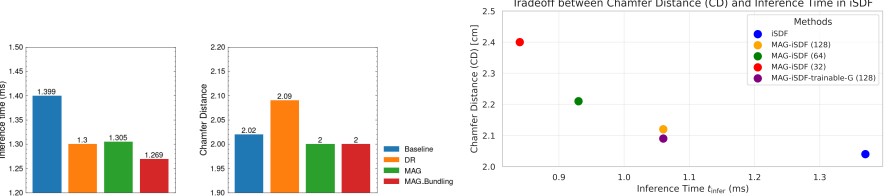

Figure 7: (Left) : Comparison of two acceleration methods: DR and MAG. The left figure shows network inference time and the right figure shows the Chamfer distance of the reconstructed mesh. MAG accelerates the network more efficiently than DR while preserving the reconstruction quality. (Right) : Trade-off between reconstruction quality and speed for various MAG models with 32, 64, 128 random features.

## 4.3 KNOWLEDGE DISTILLATION WITH MAGNITUDER

In previous sections, we demonstrate that our MAG layer can significantly accelerate the inference time of implicit neural representations. However, this acceleration requires retraining the model from scratch, a process that is both computationally intensive and time-consuming, especially for complex models like NeRF and Zip-NeRF.

To address this limitation, we introduce a layer distillation mechanism that eliminates the need for retraining on a per-scene basis. This method allows us to accelerate existing, pretrained models without having to rebuild them from the ground up. By storing the inputs and outputs of the target hidden layer in trained implicit neural representations, and then optimizing the MAG layer using a mean squared error objective, we can efficiently replicate the behavior of the original model. This optimization can be performed without the need for *backpropagation* due to the existence of a

closed-form solution (Additional details in Section B). We present additional details on distillation experiments with trainable **G** in Appendix D.5).

First, we distill a pretrained Zip-NeRF model with a MAG layer and test the distillation in *bicycle* dataset (additional details in Appendix C.3). We set the number of random features to 64, with ablations provided in Appendix D.3 (Figure 18 and Table 15). The qualitative results of the distillation are shown in Figure 9. Notably, this distillation process is lightning fast, taking only **0.12** seconds on a single RTX 4090, thanks to the closed-form solution for the input-output pairs. Since we use the same number of random features as in Section 4.2.2, we achieve the same **6**% rendering acceleration without needing to train from scratch. We show that we can recover the quality of the original NERF model while reducing the inference speeds by up to **42**% (Table 13 and Fig 8 (right)).

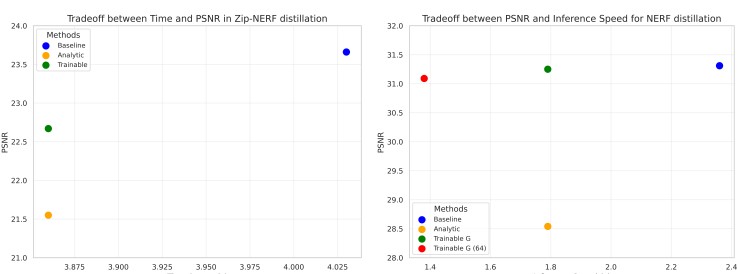

Figure 8: Distillation results comparing the analytic and trainable variants with the baseline. The analytic and trainable use 128 random features. Left : Zip-NERF, Right : NERF. Trainable **G**(64) uses 64 random features.

We also test the distillation in the last hidden layer of iSDF (Ortiz et al., 2022). In Figure 9, we visualize the reconstructed SDF and the extracted zero level-set mesh from *apt_2_nav* dataset. Our results indicate that the reconstruction quality of the distilled MAG-iSDF is comparable to that of the baseline iSDF, with a **9**% improvement in inference speed.

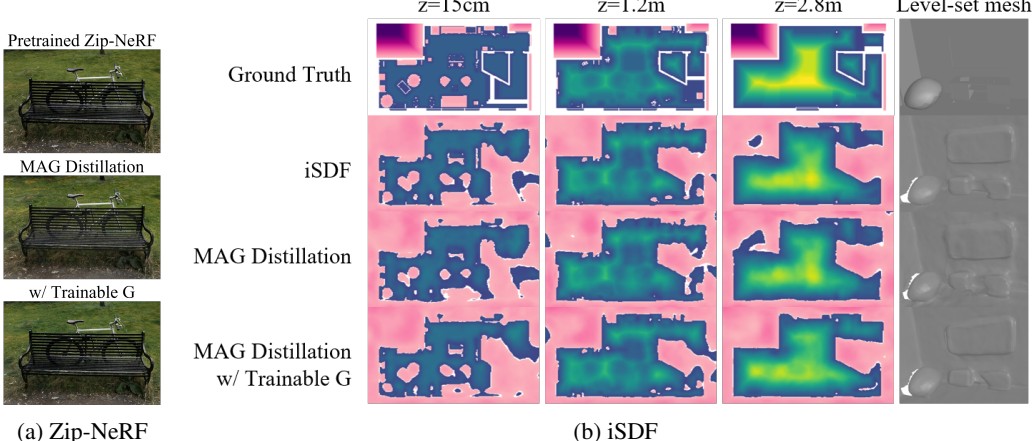

(a) Zip-NeRF            (b) iSDF

Figure 9: Visualization of reconstructed SDFs with MAG distillation. The last column shows the zoom-in rendering of zero level-set mesh.

## 5   CONCLUSION

We introduce a novel type of neural network layer, termed 'magnituders', which can efficiently approximate the computations of ReLU and Softplus linear layers. By integrating magnituders into Neural Radiance Fields (NeRF) and Signed Distance Fields (SDF) models, we achieve a reduction in training parameters while preserving expressivity. Our approach enhances inference speed and adaptability for real-time robotic applications. Additionally, the straightforward design of magnituders facilitates layer-wise knowledge distillation without requiring backpropagation. We demonstrate that this distillation process leads to efficient inference.

## ETHICS STATEMENT

This paper focuses mostly on developing novel neural network layers that can lower the computation footprint of certain feedforward layers. The experiments with NeRF and iSDF illustrate how this method can lower the inference speed while remaining competitive with the baselines. It should be noted though that even though these models are widely used, these models have considerable computational footprint and thus the corresponding carbon footprint.

## REPRODUCIBILITY STATEMENT

Details and the code pointers to reproduce each experiment are provided in Appendix C.

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

## A   COMPRESSING NETWORKS BY COMBINING LAYERS

For training NERF, we use a RF layer followed by a linear layer. In this section, we explain how this choice of architecture allows for more efficient inference. In this section, we use the PyTorch convention of describing a linear layer.

Recall the RF layer be given by the following equation :

$$\mathbf{Y}_1 = \phi_2(\mathbf{X_1})\mathbf{W_1}^\top \tag{8}$$

If the RF layer is followed by the linear layer with weight $\mathbf{W}_2$ and bias $\mathbf{b}_2$, then

$$\begin{aligned}
\mathbf{Y}_2 &= \mathbf{Y_1}\mathbf{W_2}^\top + \mathbf{b_2} \\
&= \phi_2(\mathbf{X_1})\mathbf{W_1}^\top\mathbf{W_2}^\top + \mathbf{b_2} \\
&= \phi_2(\mathbf{X_1})\hat{\mathbf{W}}_2^\top + \mathbf{b_2}
\end{aligned}$$

where $\hat{\mathbf{W}}_2 := \mathbf{W_2}\mathbf{W_1}$.

Note that the above compression works when a non-linearity is applied on the weights (as in the case of (Sehanobish et al., 2024)) or an optional bias in Equation 8.

This simple compression technique allows us to collapse the final 2 layers in our iSDF experiments allowing for more efficient inference.

## B   KNOWLEDGE DISTILLATION MADE EASY

The simple form of the RF layer makes knowledge distillation particularly simple.

To be precise, we want to train a MAG layer $\mathcal{M}$ to match the outputs of a trained FFL using MSE as the metric. If $\mathbf{x}$ (resp. $\mathbf{y}$) is the input (resp. output of a trained FFL, we want to find the weights $\hat{\mathbf{W}}$, which minimizes the following error :

$$||\mathcal{M}_\mathbf{W}(\mathbf{x}) - \mathbf{y}||_2$$

The optimal $\hat{\mathbf{W}}$ has a closed form :

$$\hat{\mathbf{W}} = (\mathbf{x'}^\top\mathbf{x'})^{-1}\mathbf{x'}^\top\mathbf{y} \tag{9}$$

where $\mathbf{x'} := \text{ReLU}(\mathbf{Gx})$, $\mathbf{G}$ is an (orthogonal) random matrix.

The above solution is clearly unique iff $\mathbf{x'}^\top\mathbf{x'}$ is invertible. In the case where $\mathbf{x'}^\top\mathbf{x'}$ is not invertible, one can replace the inverse by the Moore-Penrose pseudoinverse.

Thus the problem of distilling specific input-output pairs in the MAG layer can be solved *without backpropagation*.

## C   EXPERIMENTAL DETAILS

In this section, we provide additional details for the experiments in our main paper. Our code for the MAG layer implementation can be found `https://anonymous.4open.science/r/SNNK-NERF-2FF7`, and other experiment codes are available at `https://anonymous.4open.science/r/iSDF_MAG-2242`, `https://anonymous.4open.science/r/dnerf_MAG-69D0`, `https://anonymous.4open.science/r/zipnerf_MAG-22FF`, and `https://anonymous.4open.science/r/nerf-pytorch_MAG-425F`. Except for the core MAG-layer, all the other repos are forks of the open source repos of the respective projects.

### C.1   SYNTHETIC EXPERIMENTS

In this subsection, we provide additional experimental details for the synthetic experiments. For these experiments, 10,000 samples are drawn randomly from the range $(0, 1)$ with a dimension of

Table 6: PSNR of rendered images for NeRF reconstruction in Synthetic NeRF Mildenhall et al. (2020) dataset. Higher is better.

| Scene | chair | drums | ficus | hotdog | lego | materials | mic | ship |
|---|---|---|---|---|---|---|---|---|
| NeRF | 33.70 | 25.46 | 26.40 | 35.81 | 31.31 | 29.00 | 33.11 | 28.79 |
| MAG-NeRF | 32.76 | 24.92 | 28.11 | 35.22 | 30.14 | 28.62 | 32.42 | 28.61 |

Table 7: SSIM of rendered images for NeRF reconstruction in Synthetic NeRF Mildenhall et al. (2020) dataset. Higher is better.

| Scene | chair | drums | ficus | hotdog | lego | materials | mic | ship |
|---|---|---|---|---|---|---|---|---|
| NeRF | 0.978 | 0.930 | 0.939 | 0.979 | 0.965 | 0.956 | 0.979 | 0.870 |
| MAG-NeRF | 0.972 | 0.922 | 0.958 | 0.975 | 0.952 | 0.951 | 0.975 | 0.857 |

Table 8: LPIPS of rendered images for NeRF reconstruction in Synthetic NeRF Mildenhall et al. (2020) dataset. Lower is better.

| Scene | chair | drums | ficus | hotdog | lego | materials | mic | ship |
|---|---|---|---|---|---|---|---|---|
| NeRF | 0.013 | 0.049 | 0.048 | 0.012 | 0.017 | 0.021 | 0.020 | 0.074 |
| MAG-NeRF | 0.019 | 0.056 | 0.025 | 0.017 | 0.025 | 0.025 | 0.026 | 0.084 |

512, and are then passed through a Linear layer of shape $(512, 512)$. Non-linearity of ReLU (resp. Softplus) is applied on the outputs of the linear layer. Our aim is to show that our MAG-layer can accurately approximate these outputs.

We use the SNNK layers (Sehanobish et al., 2024) as a baseline. We train MAG and SNNK layers for 1k epochs with $8, 16, 32, 64, 128, 256, 512$ random features using the Adam optimizer (Kingma, 2014). We repeat this experiment 10 times and report the average time vs the MSE loss across the replicates. This experiment is run on free Google Colab equipped with a T4 GPU with 12Gb of RAM.

We show that our new MAG layers can accurately approximate the outputs of these linear layers while SNNKs struggle to fit the data.

## C.2 NeRF Experiments

In NeRF experiments, we use the default configuration of the Pytorch implementation of NeRF (Yen-Chen, 2020). The inference time is computed using a single NVIDIA RTX 4090 and an AMD Ryzen 7 7700 8-Core Processor. All tests are conducted on the Synthetic NeRF dataset (Mildenhall et al., 2020), and the details of our architecture choices are illustrated in Figure 3. In our experiments, we set the number of random features to 256, with additional results for the different number of random features provided in Section D.2.

We report the detailed results for each scene in Tables 6, 7 and 8 and the rendering results are in Figure 10.

## C.3 Zip-NeRF Experiments

Zip-NeRF is an extension of Mip-NeRF 360 (Barron et al., 2022), integrating explicit hash-grid for acceleration (Müller et al., 2022). The architecture consists of two shallow proposal MLPs, which help determine where samples should be placed, and a NeRF MLP, responsible for generating the final color output. We find that accelerating the proposal MLPs was challenging due to their shallow architecture and low-dimensional mapping ($6 \rightarrow 64 \rightarrow 1$). So we choose NeRF MLP as our

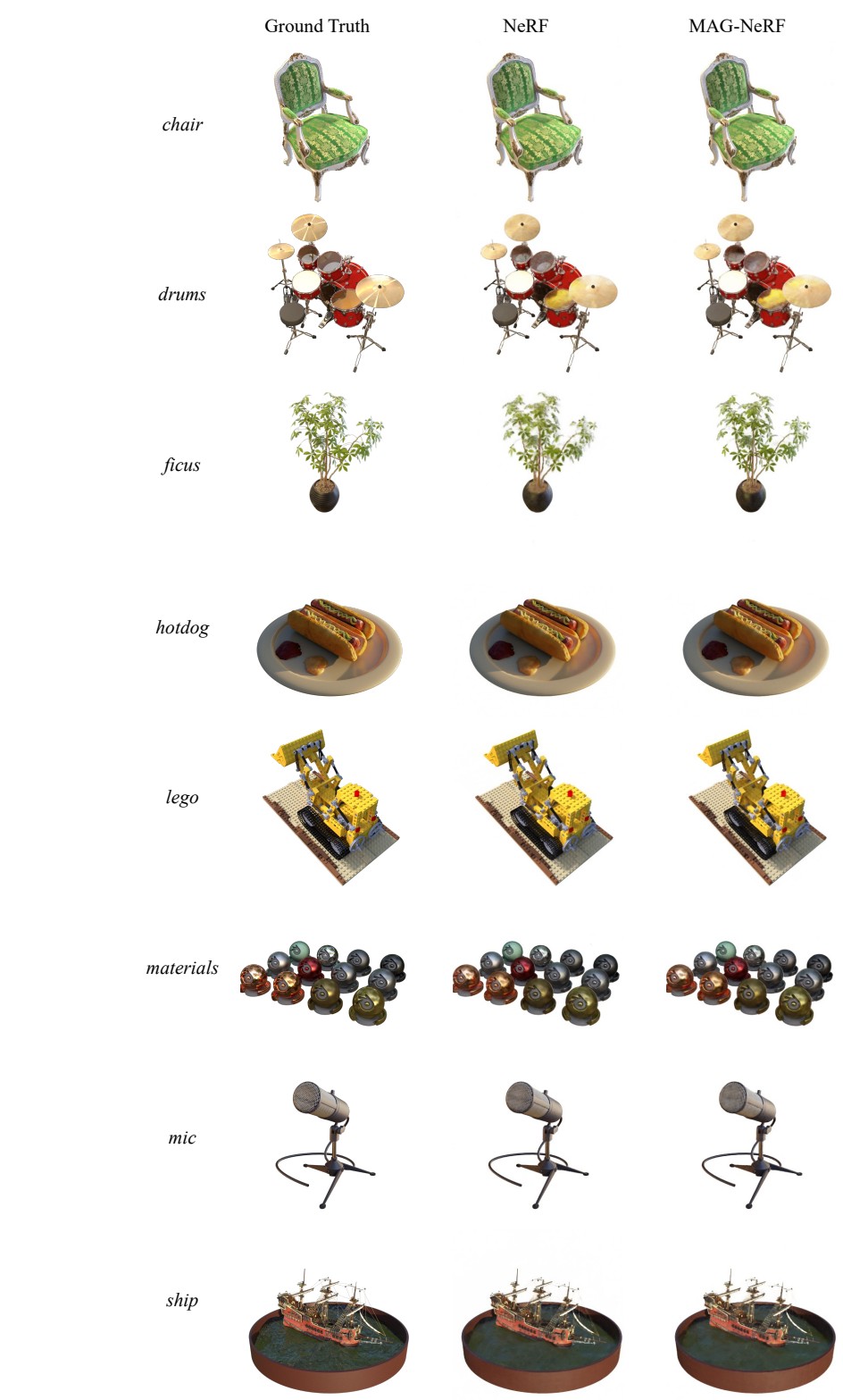

Figure 10: Rendered images from the Synthetic NeRF dataset (Mildenhall et al., 2020). For each scene, the images from left to right show the Ground Truth, NeRF (Mildenhall et al., 2020), and MAG-NeRF.

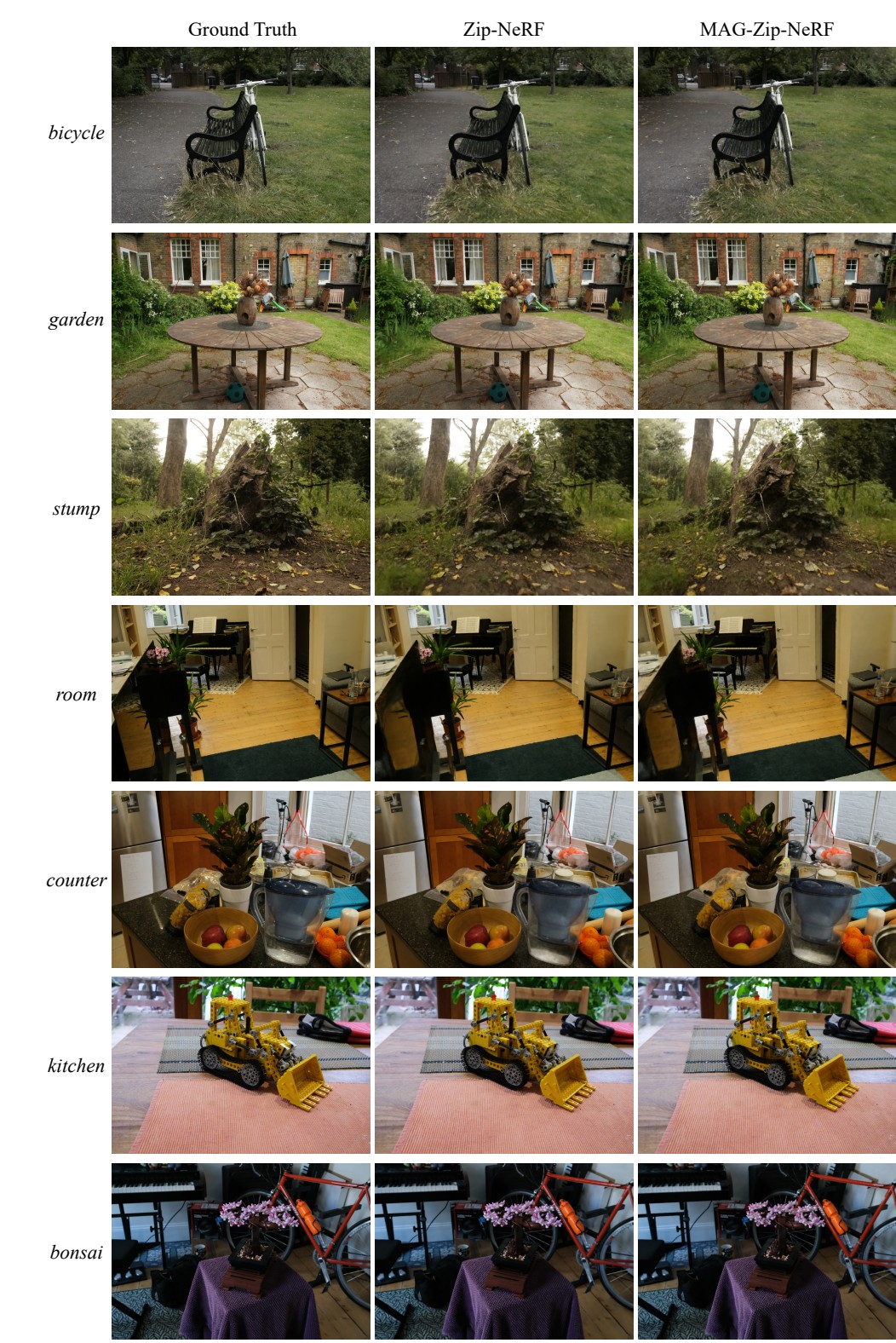

Figure 11: Rendered images from the Mip-NeRF 360 dataset (Barron et al., 2022). For each scene, the images from left to right show the Ground Truth, Zip-NeRF (Barron et al., 2023), and MAG-Zip-NeRF.

acceleration target, and we select the view-dependent part of MLP, which consists of two consecutive linear ReLU layers.

In all experiments, we use the default configuration of the PyTorch implementation of Zip-NeRF (Gu, 2023) on Mip-NeRF 360 dataset (Barron et al., 2022). The computational resources used for calculating rendering time are consistent with those in the NeRF experiments. However, due to hardware limitations, we reduce the ray batch size from 65,536 to 4,096, preventing out-of-memory issues. We set the number of random features to 64 for the MAG layer. All rendered images from Mip-NeRF 360 dataset can be shown in Figure 11.

For the distillation process, we need to get the input-output pairs from the pre-trained models. However, rendering a single image through the volume rendering process (Kajiya & Von Herzen, 1984) requires a large number of samples, which causes some memory issues if we use all of the samples for distillation. To address this, we reduce the sample size by selecting 1/16 of the images from the dataset and rendering them at 1/8 of the original training resolution. Additionally, we minimize the sample size by randomly subsampling 10% of the input-output pairs, ensuring the distillation process fits within our hardware limits and works efficiently.

## C.4 D-NeRF EXPERIMENTS

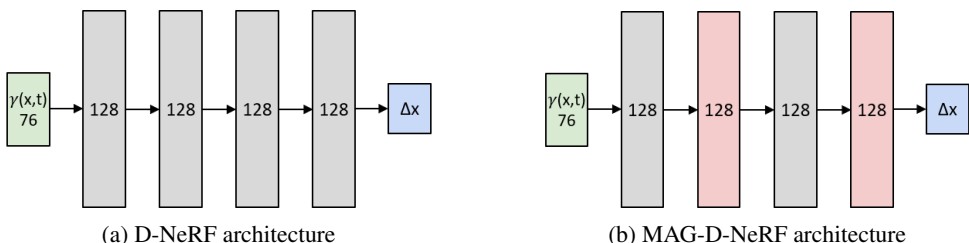

(a) D-NeRF architecture      (b) MAG-D-NeRF architecture

Figure 12: (a) Architecture of the deformation network in D-NeRF (Pumarola et al., 2021; Tang, 2022). (b) Our modified MAG-D-NeRF architecture. **Black** denotes Linear ReLU layer, **Red** is the MAG layer, and there is no nonlinear activation (ReLU) for $\Delta x$.

D-NeRF (Pumarola et al., 2021) extends NeRF to represent dynamic scenes using a deformation network. When rendering an image, the deformation network takes the position and time as inputs, producing a 3-dimensional offset that captures the current positional shift. This offset allows the ray to bend, simulating how the scene moves over time.

We use Pytorch implementation (Tang, 2022) for D-NeRF, which integrates the hash-grid (Müller et al., 2022) for further acceleration. Since it exploits the explicit grid structure, the network is shallower than the original D-NeRF. To accelerate this model, we target the deformation network, by replacing two of its layers with our MAG layers (See Figure 12 for more details).

For a fair apple-to-apple comparison between the baseline and our MAG variants, we minimize randomness by disabling certain acceleration methods in the implementation. Specifically, we use the `preload` configuration and remove other acceleration options to ensure consistency across experiments.

## C.5 MIP-NeRF 360 EXPERIMENTS

Mip-NeRF 360 Barron et al. (2022) is one of the state-of-the-art (SOTA) models for novel view synthesis. We utilize the PyTorch implementation from the NeRF-Factory (Jeong et al., 2022) codebase and reduce the $batch\_size$ from 4096 to 2048 to fit the available training memory. Inference times are measured using a single NVIDIA RTX 4090 GPU and an AMD Ryzen 7 7700 8-Core Processor. All experiments are conducted on the $360\_v2$ dataset Barron et al. (2022), with three layers replaced by our MAG layers. In our setup, the number of random features is fixed at 1024. We report the detailed results for each scene in the Tables below.

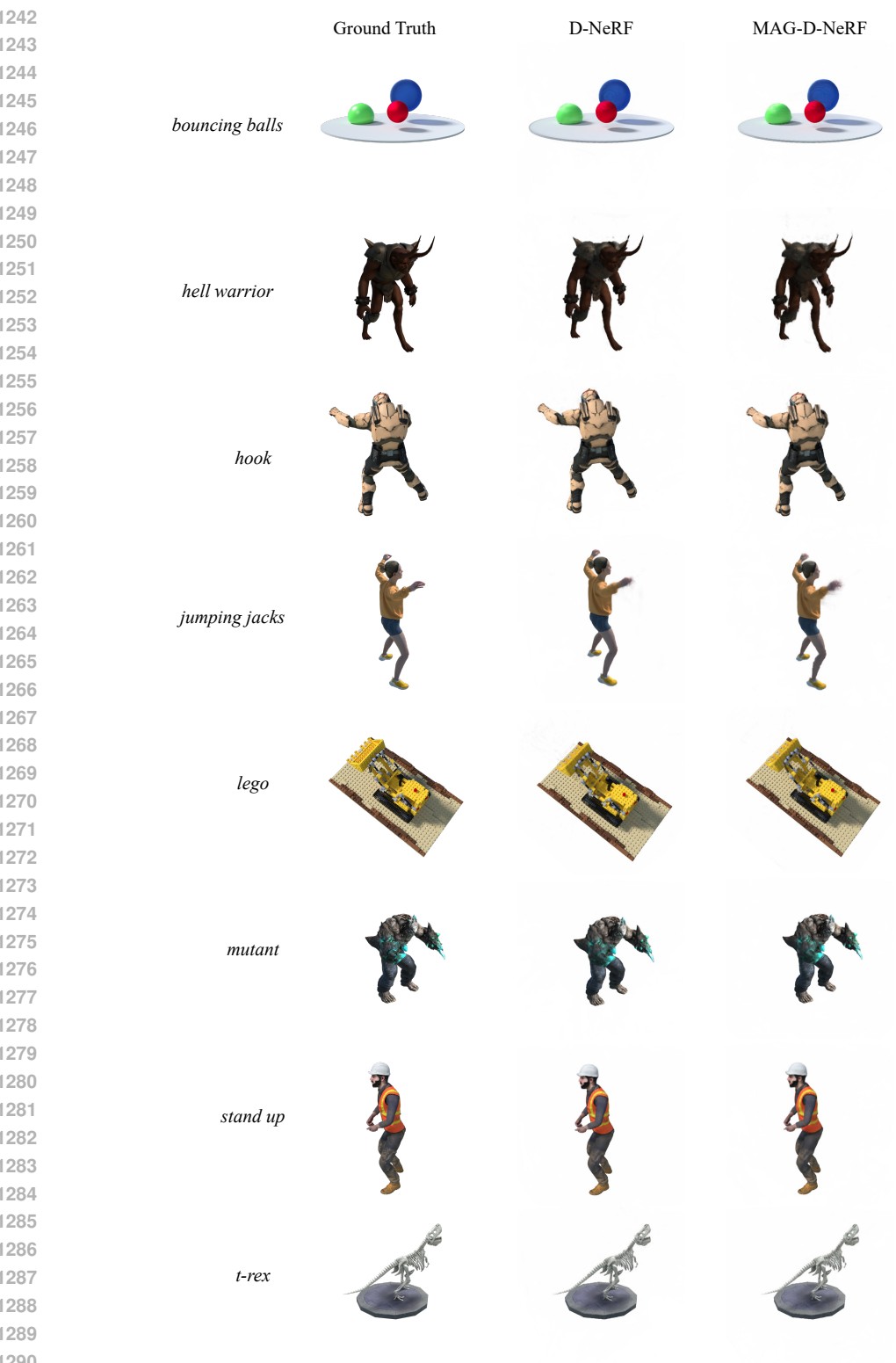

Figure 13: Rendered images from the D-NeRF dataset (Pumarola et al., 2021). For each scene, the images from left to right show the Ground Truth, D-NeRF (Pumarola et al., 2021; Tang, 2022), and MAG-D-NeRF.

Table 9: PSNR of rendered images for Mip-NeRF 360 in *360_v2* dataset. Higher is better.

| Scene | bicycle | bonsai | counter | garden | kitchen | room | stump |
|---|---|---|---|---|---|---|---|
| Mip-NeRF 360 | 22.31 | 30.59 | 27.77 | 24.65 | 29.87 | 31.18 | 24.49 |
| MAG-Mip-NeRF 360 | 22.1 | 29.99 | 27.29 | 24.30 | 29.34 | 30.6 | 24.04 |

Table 10: SSIM of rendered images for Mip-NeRF 360 in *360_v2* dataset. Higher is better.

| Scene | bicycle | bonsai | counter | garden | kitchen | room | stump |
|---|---|---|---|---|---|---|---|
| Mip-NeRF 360 | 0.466 | 0.897 | 0.809 | 0.626 | 0.879 | 0.885 | 0.569 |
| MAG-Mip-NeRF 360 | 0.435 | 0.882 | 0.793 | 0.593 | 0.859 | 0.869 | 0.538 |

Table 11: LPIPS of rendered images for Mip-NeRF 360 in *360_v2* dataset. Lower is better.

| Scene | bicycle | bonsai | counter | garden | kitchen | room | stump |
|---|---|---|---|---|---|---|---|
| Mip-NeRF 360 | 0.529 | 0.230 | 0.301 | 0.364 | 0.169 | 0.262 | 0.488 |
| MAG-Mip-NeRF 360 | 0.557 | 0.263 | 0.328 | 0.389 | 0.196 | 0.295 | 0.516 |

## C.6 iSDF Experiments

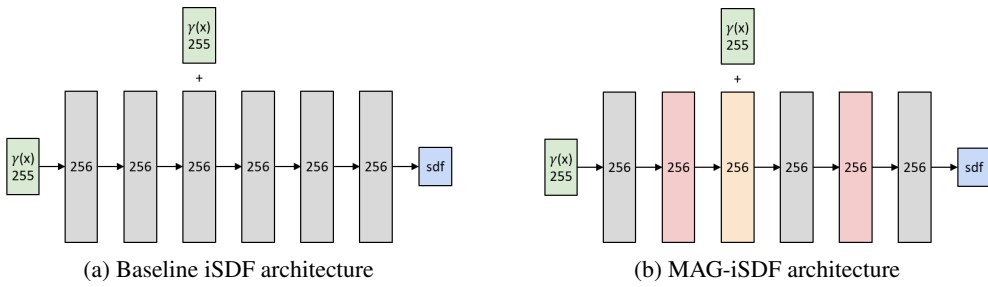

(a) Baseline iSDF architecture      (b) MAG-iSDF architecture

Figure 14: (a) Network architecture of the SDF prediction network in iSDF. (b) Our modified MAG-iSDF architecture. **Black** denotes Linear ReLU layer, **Orange** is Linear without activation, **Red** is the MAG layer, + is concatenation.

Figure 14 shows the baseline iSDF (Ortiz et al., 2022) and the modified MAG-iSDF architectures. We use six ReplicaCAD (Szot et al., 2021) scenes, following original iSDF (Ortiz et al., 2022) paper. In all experiments, we use the default configuration of public available iSDF code while replacing only the SDF network with our MAG-iSDF.

To evaluate efficiency, we report two time-related metrics: training time and inference time. However, it's important to note that, as iSDF is an incrementtal mapping module, the number of keyframes can vary between different experiment runs. Because the sample size is proportional to the number of keyframes, training time may not serve as a perfect measure of the network's efficiency. To account for this, we also report inference time, which is measured by sampling 1000 random rays and recording the time taken for a single forward pass through the network. Below we report detailed iSDF metrics and the qualitative comparison for different number of random features : 32, 64 and 128.

## C.7 Distillation Results

We show the complete distillation results for NeRF in the table below.

Table 12: Quantitative results of MAG-iSDF. RF is the number of random features, L1$_{avg}$ is calculated across all SDF levels, and L1$_{surf}$ is for ground truth surface points. $t_{train}$ refers to iteration time (including backward pass), and $t_{infer}$ measures forward pass inference time. Results are averaged across 6 ReplicaCAD (Szot et al., 2021) sequences with 3 random seeds. Our MAG-iSDF model achieves less than similar quality as the baseline while being 23% faster.

| Method | RF **G** | Distance | | | Time | |
|---|---|---|---|---|---|---|
| | | L1$_{avg}$ ($\downarrow$) [cm] | L1$_{surf}$ ($\downarrow$) [cm] | CD ($\downarrow$) [cm] | $t_{train}$ ($\downarrow$) [ms] | $t_{infer}$ ($\downarrow$) [ms] |
| iSDF | NA | 8.99 | 4.08 | 2.02 | 8.24 | 1.37 |
| | 128 | 9.62 | 4.28 | 2.12 | 7.74 | 1.06 |
| MAG-iSDF (ours) | 64 | 9.39 | 4.46 | 2.21 | 7.40 | 0.93 |
| | 32 | 9.96 | 4.38 | 2.40 | 7.13 | 0.84 |

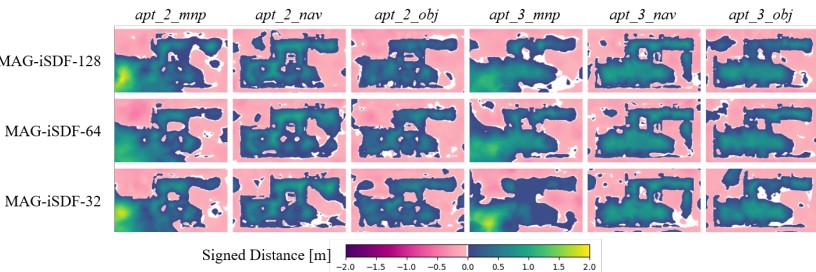

Figure 15: Visualization of reconstructed SDFs in iSDF for 6 ReplicaCAD dataset with various number of random features. Each figure shows the reconstructed SDF, following the colormap at the bottom.

Table 13: Distillation results for NeRF in *lego* dataset. Trainable **G** produces a leaner efficient model while being almost **42**% faster. Both analytic and trainable uses 128 random features except the last row which uses 64 random features.

| Method | PSNR | SSIM | LPIPS | Inference Speed (s) |
|---|---|---|---|---|
| Baseline | 31.31 | 0.965 | 0.017 | 2.36 |
| Analytic | 28.54 | 0.948 | 0.027 | 1.79 |
| Trainable **G** | 31.25 | 0.965 | 0.017 | 1.79 |
| Trainable **G** (64) | 31.09 | 0.964 | 0.017 | 1.38 |

# D  ADDITIONAL EXPERIMENTS

In this section, we present additional experiments on (a) justifying the use of ReLU over Softplus (Section D.1), (b) ablation studies (Section D.2, D.3), and (c) justify the choice of using ORF in our MAG layers.

## D.1  ReLU OVER SOFTPLUS IN iSDF

As our goal is to explore how far we can accelerate using MAG layers, we first investigate potential speed improvements in the baseline model before making modifications. We found that replacing Softplus with ReLU results in faster inference with minimal quality degradation. Specifically, this change causes a slight performance decrease for distances $s \geq 100\,\mathrm{cm}$ (see Figure 16). However, since surface information is critical for robotics tasks such as manipulation or navigation, we exploit ReLU activation in our iSDF experiments both in baseline (iSDF) and MAG-iSDF to prioritize network speed.

## D.2  EFFECT OF NUMBER OF RANDOM FEATURES IN MAG-NeRF

In this experiment, we investigate how the number of random features affects the quality of MAG-NeRF. We evaluate three scenes: *hotdog*, *materials*, and *drums*, which represent easy, medium, and

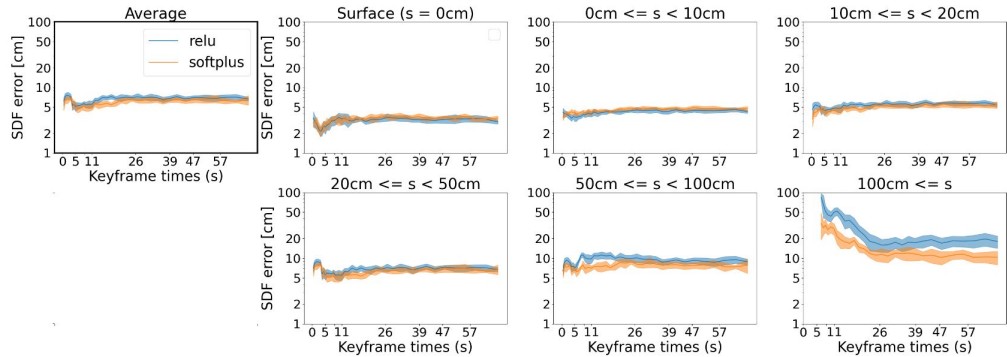

Figure 16: Error visualization comparing ground truth and reconstructed SDF using various activation functions. The first column displays the average error. The subsequent rows illustrate errors across different ranges of ground truth SDF values. In robotics scenarios such as grasping or manipulation, achieving more precise reconstruction, particularly for lower SDF values as shown in the first row, is crucial.

Table 14: Comparison between NeRF and our random feature mechanism on the *drums* dataset. The inference time refers to the approximate time for forward passes of each model to render a single 400x400 image.

| Method | RF | Rendering Quality | | | Efficiency | |
|---|---|---|---|---|---|---|
| | | PSNR ($\uparrow$) | SSIM ($\uparrow$) | LPIPS ($\downarrow$) | $t_{\text{infer}}$ ($\downarrow$) [s] | Model Size ($\downarrow$) [MB] |
| NeRF | - | 25.46 | 0.930 | 0.049 | 2.36 | 3.27 |
| MAG-NeRF | 256 | 24.92 | 0.922 | 0.056 | 1.79 | 2.27 |
| | 128 | 24.64 | 0.918 | 0.059 | 1.47 | 1.39 |
| | 64 | 24.39 | 0.913 | 0.064 | 1.32 | 0.96 |
| | 32 | 23.90 | 0.903 | 0.076 | 1.22 | 0.74 |
| | 16 | 23.17 | 0.886 | 0.098 | 1.19 | 0.63 |

Table 15: PSNR of rendered images and inference time for pretrained Zip-NeRF and MAG layer distillation using various number of random features on the *bicycle* dataset (Barron et al., 2022).

| | Pretrained | 32 | 64 | 128 | 256 | 512 |
|---|---|---|---|---|---|---|
| PSNR ($\uparrow$) | 23.66 | 20.72 | 21.06 | 21.55 | 21.72 | 22.06 |
| Inference time ($\downarrow$) (s) | 4.03 | 3.76 | 3.78 | 3.86 | 4.06 | 4.43 |

hard cases, respectively, based on rendering quality in Table 1. We test models with 16, 32, 64, 128, and 256 random features.

We present the qualitative and quantitative results of the number of random features in MAG-NeRF in Figure 17 and Table 14. Using a small number of random features, like 16, results in approximately a twofold acceleration compared to the baseline model.

### D.3 EFFECT OF NUMBER OF RANDOM FEATURES IN ZIP-NERF DISTILLATION

When distilling a linear ReLU layer into our MAG layer, the number of random features must to be set. This choice impacts the quality and acceleration trade-off. In Figure 18 and Table 15, we present an ablation study to illustrate how the number of random features affects the distillation process.

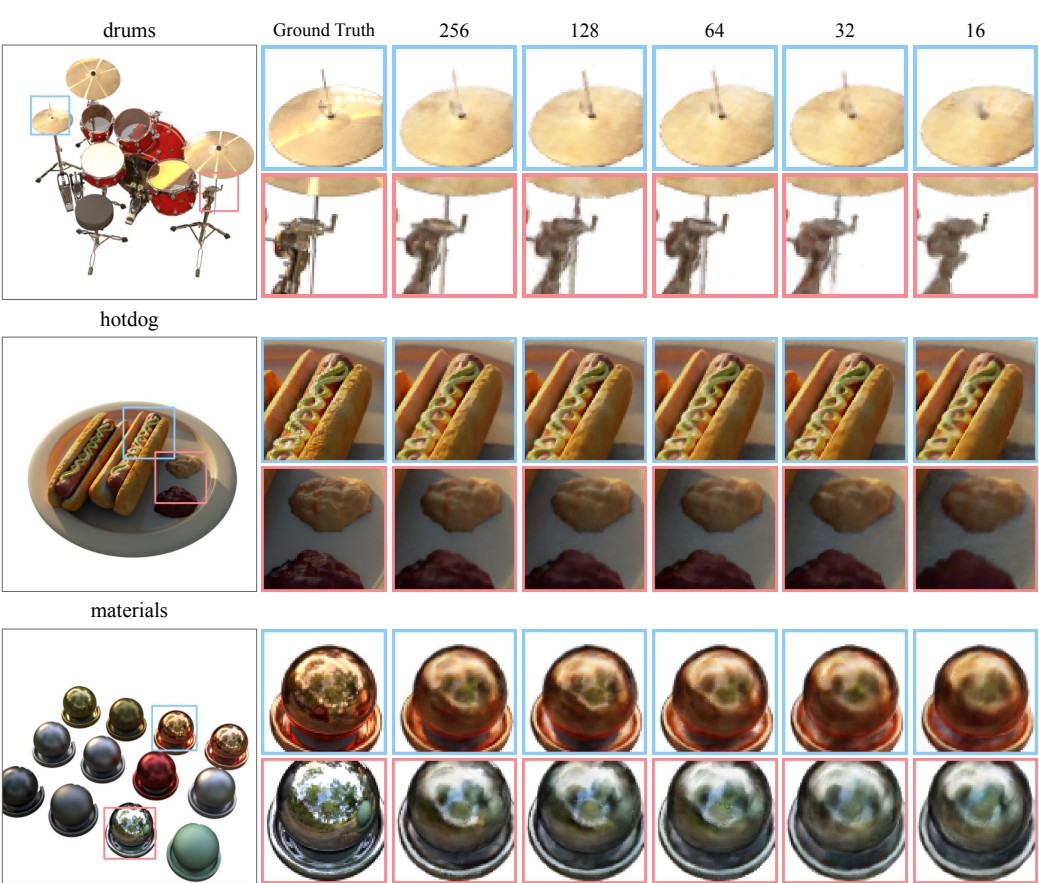

Figure 17: Comparison of rendering results with varying numbers of random features in MAG-NeRF. The numbers at the top indicate the number of random features used for each specific MAG-NeRF model. The ground truth image is on the left, while the right shows zoomed-in views of each reconstruction.

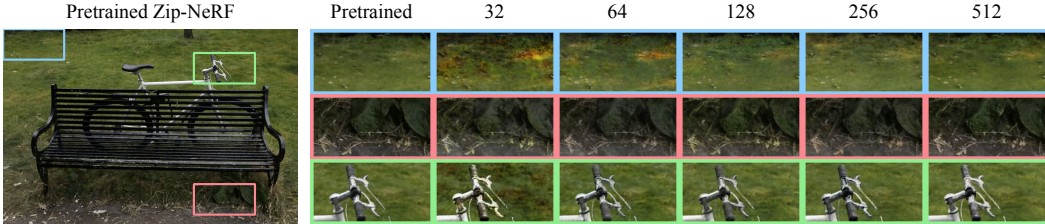

Figure 18: Comparison of rendering results with varying numbers of random features in Zip-NeRF distillation. The numbers at the top indicate the number of random features. Rendering from the target pretrained model is on the left, with zoomed-in views of each reconstruction on the right.

### D.4 ORTHOGONAL RANDOM FEATURES IN MAG

In this subsection, we show that ORFs in MAG produce better approximation quality than random projections.

### D.5 DISTILLATION EXPERIMENTS WITH TRAINABLE PROJECTION MATRIX

Recent works have shown that training the projection matrix $\mathbf{G}$ instead of using a fixed probability distribution can lead to improved results (Chowdhury et al., 2022; Zhang et al., 2024). We validate

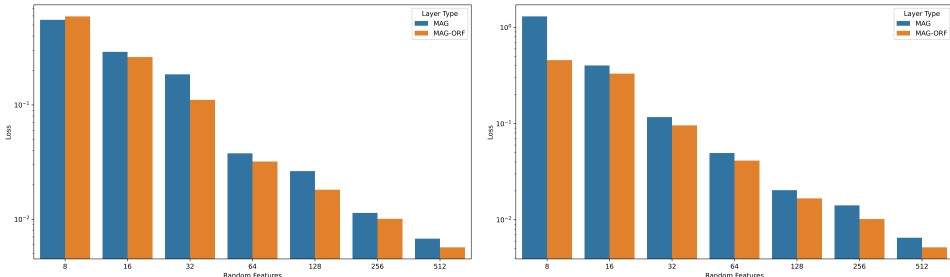

Figure 19: Comparing the approximation capability of our MAG layer with orthogonal random features vs random features. (Left) : Approximating a ReLU-Linear layer. (Right) : Approximating a Softplus-Linear layer.

this hypothesis in the context of our distillation experiments. However, optimizing the MAG layer to mimic the outputs of a FFL in a pretrained INR can not be done analytically in this case. So we use gradient descent to train this MAG layer. Table 16 shows the results for various loss functions that one can use to compare the output of MAG layer with the target FFL. We observe that the Huber loss performs better than MSE as it is robust to outliers. Adding a trainable bias improves performance, but does not add any extra latency during inference. That is why the trainable $\mathbf{G}$ with bias and Huber loss is the preferred choice for the distillation experiments.

Table 16: Performance comparison of methods based on PSNR, SSIM, and LPIPS.

| Method | Loss Function | PSNR | SSIM | LPIPS |
|---|---|---|---|---|
| Baseline | NA | 31.31 | 0.965 | 0.017 |
| Analytic (no backprop) | MSE | 28.54 | 0.948 | 0.027 |
| Trainable $\mathbf{G}$ | MSE | 31.07 | 0.964 | 0.017 |
| Trainable $\mathbf{G}$ | Huber loss | 31.11 | 0.964 | 0.017 |
| Trainable $\mathbf{G}$ + bias | MSE | 31.21 | 0.965 | 0.017 |
| Trainable $\mathbf{G}$ + bias | Huber loss | 31.25 | 0.965 | 0.017 |

## E  IMAGE CLASSIFICATION EXPERIMENTS

In this section, we showcase the usefulness of our MAG layers in image classification tasks. More specifically, we consider the problem of uptraining ViT (Dosovitskiy et al., 2021) on various downstream image classification datasets. In this setting, we replace a part of the Feed-Forward Network (FFN) block in Transformers with the MAG layer, namely the expansion layer with the GeLU activation. Note that : the MAG layer can then be combined with the following linear layer to create a single linear layer of size $[r, 768]$, where $r$ is the number of random features. This allows for a significant compression of the ViT model. In all our experiments $r = 16$. We present detailed analysis of flops in Table 17 and report the accuracy in Fig. 20 as we successively replace MLP layers starting from the top layer. To summarize, by replacing top-6 layer's MLP blocks with MAG reduces the size of the ViT model from 346 Mb to 196.85 Mb. This speeds up inference by almost 35% and has minimal impact on accuracy.

## F  A PYTORCH STYLE PSEUDO-CODE

For convenience of the readers, we present a PyTorch style pseudo-code for our MAG layer.

Table 17: Detailed analysis of parameters and flops of MAG-ViT. The number $k$ means MAG layers are inserted up to the $k$th layer starting from the top. The flops are computed for one $224 \times 224$ image.

|  | Full | 12 | 11 | 10 | 9 | 8 | 7 |
|---|---|---|---|---|---|---|---|
| # Inference parameters (millions) | 86 | 82 | 77 | 72 | 67 | 62 | 57 |
| # Inference Flops (billions) | 35 | 33 | 31 | 29 | 27 | 25 | 23 |

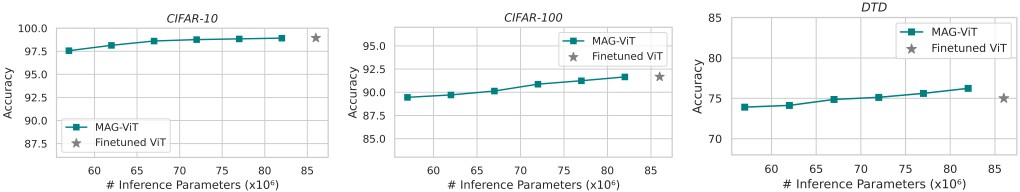

Figure 20: Accuracy vs Inference parameters trade-off plot for uptraining ViT by successively replacing MLP layers by the MAG layers from the bottom. (Left) : Accuracy vs Inference Parameters for CiFAR-10. (Middle) : Accuracy vs Inference Parameters for CiFAR-100. (Right) : Accuracy vs Inference Parameters for DTD.

---

**Algorithm 1:** PyTorch-style pseudo-code for MAG layer

---

```python
class MAG(nn.Module):
    def __init__(self, num_rf, in_dim, out_dim):
        self.projection_matrix = nn.Parameter(
            torch.randn(self.in_dim, self.num_rf),
            requires_grad=False)
        self.weight = nn.Parameter(torch.randn(out_dim, num_rf))

    def forward(self, inputs):
        # Ignoring some normalization constant
        inputs = nn.ReLU(inputs @ self.projection_matrix) # shape =
        [*, num_rf]
        outputs = inputs @ self.weight.t() # shape = [*, out_dim]
        return outputs
```

---

