# OpenReview forum: "Magnituder Layers for Implicit Neural Representations in 3D"
_ICLR.cc/2025/Conference — Submitted to ICLR 2025_

### Official Review · Reviewer_Lc1Q · 2024-10-16

**Soundness:** 3
**Presentation:** 2
**Contribution:** 2
**Rating:** 6
**Confidence:** 4

**Summary:**

This paper proposes a novel neural network layer, named MAG (magnituder), that is able to decouple the computation of inputting and weights parameters, thus accelerating the computation. The authors validate the effectiveness of MAG layers by inverse graphics (static, dynamic, and geometry) by reforming NeRF, Zip-NeRF, D-NeRF, and iSDF. The results show that the proposed MAG layers manage to accelerate the computation with small performance drop.

**Strengths:**

- The paper attempts to solve the problem of computational costs in the filed of Implicit Neural Representation, which is clearly motivated and significant for making INRs more scalable.
- The proposed MAG layers follow a plug-and-play fashion, making the integration into most of MLP-based INRs easy and fast, which would be greatly valued by the users. The core idea of simplifying the computation of FC layer is interesting.
- The experiments do showcase certain effectiveness of the MAG layers in terms of reducing the inference time at a cost of some performance.

**Weaknesses:**

- The elaboration of the adopted methods (Sec.3) is somehow not sufficient to me. The authors are encouraged to include another illustration or pseudo code to better describe their method.
- Though the connection between MAG and SNNK is covered, such strategy for simplification still requires comparison between other well-known alternatives, e.g., distillation, pruning, and also separable convolution.
- The overall writing of this manuscript is not informative. For example, Sec.2 is a comprehensive but meaningless review of the applications of INRs, which is not closely related to the scope of the manuscript. Instead, the authors are supposed to include more strategies for simplifying or accelerating INRs.
- The quantitative results are not satisfactory enough to match the claimed contribution. A universal decline of performance can be witnessed across basically all use cases under all evaluation metrics while the acceleration is just marginal. Besides, the measure of computational costs should cover more aspects in addition to just time, which can be easily affected by complicated factors.

**Questions:**

Considering the aforementioned strengths and weaknesses, I tend to give a reject as the initial rating. The authors are also encouraged to refer to the weaknesses for rebuttal.
- I personally consider the effectiveness of MAG can be explained intuitively by dictionary learning. The $W$ can be seem as a learnable dictionary to model a sparse mapping from $x$ to the next layer. Can the authors provide more insights towards the mechanism of MAG? Also, is there any potential connection between MAG and separable convolution which decouples channels from tensors.
- Zip-NeRF is known for its large memory consumption (referring to NeRF-baselines). Is it possible to reduce the massive overheads by applying MAG layers or the improvement can only be time-wise?
- How does the proposed method perform compared to common acceleration techniques? The aim of this question is to confirm the contribution of MAG is not trivial and not easily achieved by other simple operations.
- How does the MAG-inserted MLP converge? Will there be a different tendency compared to vanilla MLP?

---

> ### Author Response · Authors · 2024-11-19
>
> > *Zip-NeRF is known for its large memory consumption (referring to NeRF-baselines). Is it possible to reduce the massive overheads by applying MAG layers or the improvement can only be time-wise?*
>
> Thank you for this question. The large memory consumption in Zip-NeRF primarily stems from its explicit grid structure, which MAG layers alone cannot reduce. This is precisely where fully implicit models, such as vanilla NeRF, become advantageous under strict memory constraints. In such scenarios, MAG layers serve as an effective and easy-to-implement method for accelerating inference without adding to memory overhead.
>
>
> > *Quantitative results are not satisfactory enough*
>
> We would like to clarify that we get gains for not only NERF but for iSDF, MIP-NERF as well when we train them from scratch. For NERF, we get **24 %** speedups while suffering only **1%** drop in quality. Similarly, for iSDF we get **23%** gain (see the trade-off plot in Fig 7 (right)). To get more quality gains, we borrowed ideas from recent works on random feature mechanisms.
>
> [1,2] have shown improved performance when the projection matrix $\mathbf{G}$ can be trained instead of sampled from a fixed distribution. We show that it is indeed true in our case as well. Note that this incurs no additional cost in inference. In all cases, including NERF, D-NERF, MipNERF, ZipNERF, iSDF, ViT we show that using trainable $\mathbf{G}$, our performance drop is minimal and we have updated all the tables in the paper with the new updated results. For convenience, we summarize them in the table below :
>
> | Model      | Delta in Performance (%) | Delta in Inference Speed (%) |
> |------------|---------------------------|-------------------------------|
> | NERF       | -1                         | +24                            |
> | iSDF       | -4                         | +23                            |
> | D-NERF     | - <1                        | +6                             |
> | Zip-NERF   | -1                         | +7                             |
> | ViT        | -2                         | +35                            |
> | Mip-NERF   |       -2                    |                 +18              |
>
> Thus our method becomes quite powerful in INRs and in models where there are a lot of MLP layers. Furthermore these gains are higher than some existing published work (for ex: see [3] where the inference gain is around 18% with a quality drop of 3%).
>
> Our main focus in this work is in INRs where one has a lot of MLPs. For other shallower architectures, our methods provide gains but the gains are small compared to INRs as we have a limited room to inject our MAG layers.
>
> Finally, our distillation method is actually a real world test case of Q1 where the distributions are coming from pretrained models. Our MAG-layers can approximate complex outputs well enough that they can simply replace FFLs, thus justifying the design choice of $f$. Thus this simple layer wise distillation method is supposed to be a test-time inference speed up method where one does not have access to a lot of resources. This distillation takes less than a second and very limited GPU resources to train. Our new results on layer-wise distillation can actually maintain quality while reducing inference speeds (see Fig 8 and 9).  This method is designed to complement other methods like pruning and quantization to speed up inference.
>
> Finally since the number of random features is smaller than the dimension of the models, the memory footprint for training these models is less than the base models. Furthermore, the design of these layers allow for combining the MAG-layer with the subsequent linear linear effectively creating a single linear layer (please see the details in Appendix A) which further reduces the memory footprint during inference which is beneficial for deployment in low resource scenarios like edge devices.
>
> [1] On Learning the Transformer Kernel. Chowdhury et al. 2022
>
> [2] The Hedgehog & the Porcupine: Expressive Linear Attentions with Softmax Mimicry. Zhang et al. ICLR 2024
>
> [3] CoordX: Accelerating Implicit Neural Representation with a Split MLP Architecture. Liang et al. ICLR 2022
>
> > *Converge of MAG-MLP vs regular MLP*
>
> Thank you for the question. We do not see any issues in convergence for the MAG-MLP. We simply use ADAM as an optimizer and the auto-differentiation in PyTorch to train these models.
>
> > *Pseudo-code for our algorithm*
>
> We have added a PyTorch style pseudo-code for our algorithm in Appendix F.

---

> > ### Author Response · Authors · 2024-11-19
> >
> > > *Comparison with other strategies*
> >
> > As explained in our work, our strategy is completely complementary to pruning and quantization techniques and can be combined with such. Our main goal in this work is to create a new type of a neural network layer that can mimic the computation of certain feedforward layers (FFL). The disentangling of weights and inputs is a powerful method as it allows a MAG to be combined with **any** subsequent FFL, essentially collapsing a 2-layer network into a single layer. The design of this layer also allows one to compute the optimal weights for loss functions like MSE loss without the need for backpropagation. This line of work in viewing a FFL as a kernel method is quite recent and is only introduced in [1].
> >
> > We have experimented with layer-wise distillation. We view this as a real world test case of Q1 where the distributions are coming from pretrained models. Our MAG-layers can approximate complex outputs well enough that they can simply replace FFLs, thus justifying the design choice of $f$. Thus this simple layer wise distillation method is supposed to be a test-time inference speed up method where one does not have access to a lot of resources.
> >
> > [1] Scalable Neural Network Kernels. Sehanobish et al. ICLR 2024
> >
> > > *Other speed up techniques like reducing parameter count*
> >
> > Thank you for proposing this experiment. We added these results in Fig 5. We show that our models offer similar or higher reconstruction quality than the models with the same number of training parameters while being consistently faster. The speed-ups are achieved due to the compression trick as explained in Appendix which allows us to combine any subsequent FFL. This results in effectively converting a 2-layer neural network into a single layer. For NERF, we observe that reducing the size of the model to match our parameter count results in **2 %** degradation in quality, while still being almost **5 %** slower.
> >
> > Furthermore, we also show that reducing the layer size in iSDF to match the number of training parameters in our proposed MAG-iSDF results in a **4.5%** performance decrease compared to MAG-iSDF. The smaller iSDF is also **2%** slower than our MAG-iSDF due to our novel compression trick. (see Fig 7).
> >
> > Thus simple speed-ups techniques do not work quite as well as our MAG-layers which can retain quality as well as achieve speedups.
> >
> > > *MAG layers for other tasks*
> >
> > Since MAG layers are a general neural network, they can also be used in Transformers which have a lot of MLP layers. We apply MAG-layers for the task of image classification using Vision Transformer (ViT). In this setting, we replace a part of the Feed-Forward Network (FFN) block in ViT with the MAG layer, namely the expansion layer with the GeLU activation. Note that : the MAG layer can then be combined with the following linear layer to create a single linear layer of size $[r, 768]$, where $r$ is the number of random features. This allows for a significant compression of the ViT model. To summarize our findings : by replacing top-6 layer’s MLP blocks with MAG reduces the size of the ViT model from 346 Mb to 196.85 Mb. This speeds up inference by almost $\mathbf{35}$% and has minimal impact on accuracy (less than 2%) . We present more details in Appendix E Image Classification Experiments. For convenience, we also show the results comparing inference parameters (in millions) vs accuracy in the table below :
> >
> > | Dataset      | 86  (Full fine-tuning)   | 82     | 77     | 72     | 67     | 62     | 57     |
> > |--------------|--------|--------|--------|--------|--------|--------|--------|
> > | CIFAR-10     | 98.95  | 98.92  | 98.84  | 98.76  | 98.61  | 98.14  | 97.56  |
> > | CIFAR-100    | 91.67  | 91.65  | 91.24  | 90.87  | 90.13  | 89.71  | 89.45  |
> > | DTD    | 75.0  | 76.23  | 75.61  | 75.12  | 74.86  | 74.31  | 73.94  |
> >
> > >  *General Significance of our work*
> >
> > We would like to point out the following novelties of our work :
> >
> > - Thinking of FFL computation as a kernel method is a new method and it comes with theoretical guarantees with principled variance reduction techniques.
> > - Reimagining the computation as a kernel method allows us to combine a MAG linear with **any** subsequent FFL, effectively compressing a 2-layer network into a single layer.
> > - This new method is completely orthogonal to all techniques that are used to commonly speed computation of FFLs or INRs in particular, and so can be combined with them.
> > - This layer is general in nature and can actually be used to reduce the computation burden of models where there are a lot of FFLs.
> > - The number of random features is a hyperparameter and can be used to trade-off quality vs time or memory.
> > - **Expressiveness:** Our MAG mechanism is not contained within the regular FFL framework,  and is capable of modeling relationships beyond those of standard FFLs.

---

> > > ### Author Response · Authors · 2024-11-19
> > >
> > > > *include more strategies for simplifying or accelerating INRs.*
> > >
> > > In our related works, we have included strategies for accelerating NERFs like using various explicit representations or caching or dividing a scene into smaller blocks. In particular, we have cited **23** papers that use these tricks to accelerate NERFs (see lines 91-97). For SDFs, we have cited **10** works that use various strategies like using explicit grid structures with implicit SDF, high quality triangular meshes and populating thin shells near level zero sets (see lines 112-116). We have also mentioned some of these techniques in the introduction (see lines 58-61). Our method is simply orthogonal to these techniques as we try to reduce the computation complexity of a given feed forward layer.
> > >
> > > To the best of our knowledge, most works in accelerating INRs follow a similar strategy of using explicit structures with implicit representations.
> > >
> > > We tried to do our due diligence and do a thorough literature review on various acceleration techniques for INRs. We apologize if we missed any important and relevant strategies. Please let us know if we missed any relevant methods and we would be happy to add it in the manuscript.
> > >
> > > > *effectiveness of MAG can be explained intuitively by dictionary learning. Connections with separable convolutions*
> > >
> > > We do not think this method does not learn or leverage any sparse representations so the connection with dictionary learning is not clear. The projection matrix on the inputs is not learned (in general) but simply sampled from some probability distribution. We sincerely appreciate it if the reviewer can clarify the link between MAG and dictionary learning.
> > >
> > > We think of this method as a kernel method which learns the relationship between weights and (transformed/projected)  inputs. There is a general recipe in [1] where many FFL computations can be thought of as kernel methods. But both MAG and SNNKs are more general than standard FFLs and can encode computations beyond FFLs.
> > >
> > > Separable convolution is a method that breaks down a convolution operator. However, separable convolutions can not disentangle the weights and inputs, for ex given a nonlinearity $f$ applied point-wise : $f(\mathbf{W}\mathbf{x})$, and if $\mathbf{W}$ is the convolution operator, it can not write it as :
> > > $f(\mathbf{W}\mathbf{x}) \sim \mathbf{W}_ 1 \mathbf{x}_1$, due to the existence of the nonlinearity $f$. In our works, we view the operation : $f(\mathbf{W}\mathbf{x}) \sim \phi(\mathbf{W}) \psi(\mathbf{x})$, for random feature mechanisms $\phi$ and $\psi$. One of the main motivations to disentangle the weights and inputs (essentially creating a simple linear layer) is that this layer can be combined with **any** subsequent linear layer essentially converting a 2 layer network into a simple layer.

---

> > > > ### Comment · Reviewer_Lc1Q · 2024-11-20
> > > >
> > > > Thanks to the authors for providing such an informative rebuttal for clarification.
> > > >
> > > > I personally consider MAG as a low-rank decomposition (where the reduction of parameters roots in) of $f(\mathbf{Wx})$, where separable convolutions and dictionary learning are two alternatives. However, I am now convinced by the response that MAG should be considered as a kernel method.
> > > >
> > > > The quantitative comparisons of performance drop and speed gain are especially appreciated, too.

---

> ### Author Response · Authors · 2024-11-20
>
> Dear Reviewer,
>
>  Thank you for your response. We will add a discussion around low rank decompositions with focus around dictionary learning and separable convolutions. Please do not hesitate to reach out if you have any questions.
>
> If our rebuttal has successfully answered your concerns, we would kindly request the Reviewer to adjust their scores.
>
> Sincerely,
>
> The Authors.

---

> > ### Comment · Reviewer_Lc1Q · 2024-11-26
> > **About final rating**
> >
> > The authors manage to solve most of my major concerns and I would like to raise the final rating to a marginal accept.

---

> > > ### Author Response · Authors · 2024-11-27
> > >
> > > Dear Reviewer,
> > >
> > >  Thank you for updating your score. We greatly appreciate your feedback and comments, as they contribute to making our paper better.
> > >
> > > Thank you,
> > >
> > > The Authors.

---

### Official Review · Reviewer_oXcM · 2024-10-31

**Soundness:** 2
**Presentation:** 3
**Contribution:** 2
**Rating:** 5
**Confidence:** 3

**Summary:**

This work presents a new neural network layer, magnituder (MAG), to improve the efficiency of MLP models used on implicit neural representations (NeRF, SDF, etc.). Magnituder disentangles the weights and inputs of MLP layers by injecting a mapping function operating on the magnitude of the input. The design of magnituder is supported by the analysis of the approximation of kernel methods. Magnituder does provide improvements in inference speed and model size but not significant.

**Strengths:**

* The proposed method is well-analyzed through kernel estimation methods and the theoretical comparison with the prior approach SNNK. This makes the proposed method technically sound.
* The authors apply MAG-layers in multiple applications, which is appreciated.

**Weaknesses:**

Although the paper introduces an interesting new layer with some random features to speed up MLP models used in implicit neural representations, the improvements from their results are not that exciting (compared to other NeRF acceleration works in recent years). Other than the original NeRF, the proposed MAG layers can only provide less than 10% inference speed improvements. Besides, this marginal inference speedup is also at the cost of a quality drop.

Given the current trend towards neural 3D representations that use very few and tiny MLP layers or even no MLP layers (e.g., InstantNGP, 3DGS), the benefits of applying MAG-layers to such architectures would likely be very insignificant.

It would be better to see if the MAG-layers could be beneficial for other applications that use a lot of MLP layers, not just on implicit neural representations.

**Questions:**

1. As MAG is a lossy acceleration method, you can also reduce the number of parameters used in baseline NeRF MLP to make it faster but with a slightly lower quality. For example in Fig 3., you can reduce the hidden feature size in the baseline NeRF to match the random feature size in the corresponding red block in the MAG-NeRF. In this case, does MAG-NeRF still perform better in quality and speed?
2. How about the speedup on the training time?

---

> ### Author Response · Authors · 2024-11-19
>
> > *Given the current trend towards neural 3D representations that use very few and tiny MLP layers or even no MLP layers (e.g., InstantNGP, 3DGS), the benefits of applying MAG-layers to such architectures would likely be very insignificant.*
>
> While methods like Instant-NGP, and 3DGS achieve high speed-ups, they rely on hybrid or explicit representations that increase memory usage. Thus fully implicit methods are useful in scenarios with limited memory resources like edge devices. Both SDFs and NERFs find applications in various robotics tasks. We present two examples below :
>
> - Truncated Signed Distance Fields were recently successfully applied to create 3D-fields for Robotic manipulation (re-arrangement problem) (applying large VLMs in zero-shot fashion + MPC on learned dynamics) [1] .
> - In [2], the authors used a NERF to create reconstructions of an office environment.
>
> Thus our main focus in this work is on INRs. For models with a large number of MLPs, like NERF, iSDF, ViT and MIP-NERF, we consistently get almost $20$% gains in inference speeds with minimal drop in quality (about 2%).
>
> [1] $D^3$ Fields: Dynamic 3D Descriptor Fields for Zero-Shot Generalizable Rearrangement. Wang et al. CoRL 2024
>
> [2] Mobility VLA: Multimodal Instruction Navigation with Long-Context VLMs and Topological Graphs. Chiang et al.  CoRL 2024
>
>
> > *MAG Layers for other applications*
>
> Thank you for this question.
>
> We apply MAG-layers for the task of image classification using Vision Transformer (ViT). In this setting, we replace a part of the Feed-Forward Network (FFN) block in ViT with the MAG layer, namely the expansion layer with the GeLU activation. Note that : the MAG layer can then be combined with the following linear layer to create a single linear layer of size $[r, 768]$, where $r$ is the number of random features. This allows for a significant compression of the ViT model. To summarize our findings : by replacing top-6 layer’s MLP blocks with MAG reduces the size of the ViT model from 346 Mb to 196.85 Mb. This speeds up inference by almost $\mathbf{35}$% and has minimal impact on accuracy (less than 2%) . We present more details in Appendix E Image Classification Experiments. For convenience, we also show the results comparing inference parameters (in millions) vs accuracy in the table below :
>
> | Dataset      | 86 (Full fine-tuning)    | 82     | 77     | 72     | 67     | 62     | 57     |
> |--------------|--------|--------|--------|--------|--------|--------|--------|
> | CIFAR-10     | 98.95  | 98.92  | 98.84  | 98.76  | 98.61  | 98.14  | 97.56  |
> | CIFAR-100    | 91.67  | 91.65  | 91.24  | 90.87  | 90.13  | 89.71  | 89.45  |
> | DTD    | 75.0  | 76.23  | 75.61  | 75.12  | 74.86  | 74.31  | 73.94  |
>
> > *Speed up during training*
>
> Thank you for this question. One of the benefits of using a smaller number of random features is that it shrinks the memory footprint during training. We compare the training time for iSDF where our method achieves **10%** improvement in training time.
>
> > *Reducing training parameters of baseline NERF*
>
> Thank you for proposing this experiment. We added these results in Fig 5. We show that our models offer similar or higher reconstruction quality than the models with the same number of training parameters while being consistently faster. The speed-ups are achieved due to the compression trick as explained in Appendix A which allows us to combine any subsequent FFL. This results in effectively converting a 2-layer neural network into a single layer. For NERF, we observe that reducing the size of the model to match our parameter count results in **2 %** degradation in quality, while still being almost **5 %** slower.
>
> Furthermore, we also show that reducing the layer size in iSDF to match the number of training parameters in our proposed MAG-iSDF results in a **4.5%** performance decrease compared to MAG-iSDF. The smaller iSDF is also **2%** slower than our MAG-iSDF due to our novel compression trick. (see Fig 7).
>
> Thus simple speed-ups techniques do not work quite as well as our MAG-layers which can retain quality as well as achieve speedups.

---

> > ### Author Response · Authors · 2024-11-19
> >
> > > *Other than the original NeRF, the proposed MAG layers can only provide less than 10% inference speed improvements.*
> >
> > We would like to clarify that we get gains for not only NERF but for iSDF as well when we train them from scratch. For iSDF we get **23%** gain (see the trade-off plot in Fig 7 (right)). To get more quality gains, we borrowed ideas from recent works on random feature mechanisms.
> >
> > [1,2] have shown improved performance when the projection matrix $\mathbf{G}$ can be trained instead of sampled from a fixed distribution. We show that it is indeed true in our case as well. Note that this incurs no additional cost in inference. We re-ran the experiments in our paper with this variant. In all cases, including NERF, D-NERF, MipNERF, ZipNERF, iSDF, ViT we show that using trainable $\mathbf{G}$, our performance drop is minimal and have updated all the tables in our paper with the new results. For convenience, we summarize them in the table below :
> >
> > | Model      | Delta in Performance (%) | Delta in Inference Speed (%) |
> > |------------|---------------------------|-------------------------------|
> > | NERF       | -1                         | +24                            |
> > | iSDF       | -4                         | +23                            |
> > | D-NERF     | - <1                        | +6                             |
> > | Zip-NERF   | -1                         | +7                             |
> > | ViT        | -2                         | +35                            |
> > | Mip-NERF   |      -2                     |              +18                 |
> >
> > Thus our method becomes quite powerful in INRs and in models where there are a lot of MLP layers. Furthermore, these gains for NERF are higher than some existing published work (for ex: see [3] where the inference gain is around 18% with a quality drop of 3%).
> >
> > Our main focus in this work is in INRs where one has a lot of MLPs. For other shallower architectures, our methods provide gains but the gains are small compared to INRs as we have a limited room to inject our MAG layers.
> >
> > Finally, our distillation method is actually a real world test case of Q1 where the distributions are coming from pretrained models. Our MAG-layers can approximate complex outputs well enough that they can simply replace FFLs, thus justifying the design choice of $f$. Thus this simple layer wise distillation method is supposed to be a test-time inference speed up method where one does not have access to a lot of resources. This distillation takes less than a second and very limited GPU resources to train. Our new results on layer-wise distillation can actually maintain quality while reducing inference speeds (see Fig 8 and 9).  This method is designed to complement other methods like pruning and quantization to speed up inference.
> >
> > [1] On Learning the Transformer Kernel. Chowdhury et al. 2022
> >
> > [2] The Hedgehog & the Porcupine: Expressive Linear Attentions with Softmax Mimicry. Zhang et al. ICLR 2024
> >
> > [3] CoordX: Accelerating Implicit Neural Representation with a Split MLP Architecture. Liang et al. ICLR 2022
> >
> > > *General Significance of our work*
> >
> > We would like to point out the following novelties of our work :
> >
> > - Thinking of FFL computation as a kernel method is a new method and it comes with theoretical guarantees with principled variance reduction techniques.
> > - Reimagining the computation as a kernel method allows us to combine a MAG linear with **any** subsequent FFL, effectively compressing a 2-layer network into a single layer.
> > - This new method is completely orthogonal to all techniques that are used to commonly speed computation of FFLs or INRs in particular, and so can be combined with them.
> > - This layer is general in nature and can actually be used to reduce the computation burden of models where there are a lot of FFLs.
> > - The number of random features is a hyperparameter and can be used to trade-off quality vs time or memory.
> > - **Expressiveness:** Our MAG mechanism is not contained within the regular FFL framework, and is capable of modeling relationships beyond those of standard FFLs.

---

> ### Comment · Reviewer_oXcM · 2024-11-23
>
> I went through the author's responses. I appreciate the additional efforts made by the authors during the rebuttal.
> Below are my responses:
>
> 1. The authors made additional experiments to show that MAG layer indeed performs better than simply the number of parameters in the layers. The results on NeRF & D-NeRF, the dense MLPs, show the superiority of MAG layers. However, on Zip-NeRF, MAG-layer has a slight quality drop compared to the reduced parameter version. These results indicate that the MAG layer can be a better way to reduce parameter size and improve the inference speed, for models with dense MLP layers. I acknowledge this contribution claimed in the paper.
>
> 2. I pretty much agree with Reviewer TtzD's [comments](https://openreview.net/forum?id=uswS6tUCN2&noteId=QRj7KTnL74). The MAG-layer should not only positioned as some acceleration technique for 3D representation. The proposed method is more generalizable and can be applied to models beyond INR, for example, like ViT, LLM, etc.
>
> Given the current status of this submission, I will keep my rating score around the borderline of acceptance.

---

> > ### Author Response · Authors · 2024-11-23
> >
> > Dear Reviewer,
> >
> > We sincerely thank you for your feedback and comments.
> >
> >  Best,
> >
> >  The Authors.

---

### Official Review · Reviewer_TtzD · 2024-11-02

**Soundness:** 2
**Presentation:** 2
**Contribution:** 2
**Rating:** 5
**Confidence:** 3

**Summary:**

This paper proposes a novel neural network layer called the "Magnituder," designed to reduce the number of training parameters in implicit 3D representation models like NeRF and SDF. The proposed method can also be integrated into pretrained NeRF models without additional optimization. Extensive experiments on multiple datasets and tasks demonstrate that this approach achieves faster inference speed and reduced model size.

**Strengths:**

1. This paper performs extensive experiments to thoroughly demonstrate the capability of the proposed method.
2. The proposed method improves inference speed and reduces the model size of implicit 3D representations.

**Weaknesses:**

1. While the proposed neural network achieves speedup and compression, it does so at the cost of rendering quality. This seems more like a trade-off than a true improvement, as there are other methods in 3D representation, such as 3D Gaussian Splatting, Instant-NGP, and Neus2, that significantly increase inference speed (up to 10x) without compromising rendering quality. This limitation diminishes the significance of the contribution.
2. A simple baseline may achieve similar effects to the proposed method by reducing the number of layers in the original MLP. This approach could also reduce parameters and increase inference speed, potentially affecting rendering quality. The authors are encouraged to include this comparison.
3. The proposed Magnituder layer lacks any designs specifically for 3D tasks. As a general layer that can be integrated into MLPs, it could also be applied to other tasks like classification, or in models with MLP components like ResNet or Transformer. To fully demonstrate the benefits of this layer, it would be valuable to test it on other tasks as well.

**Questions:**

Please see weaknesses.

---

> ### Author Response · Authors · 2024-11-19
>
> >  *Other methods in 3D representation, such as 3D Gaussian Splatting, Instant-NGP, and Neus2, that significantly increase inference speed (up to 10x) without compromising rendering quality.*
>
> While methods like 3D-Gaussian Splatting (3D-GS), Instant-NGP, and NeuS2 achieve high speed-ups, they rely on hybrid or explicit representations that increase memory usage. This high memory usage is not well-suited for scenarios with limited resources like on edge devices, where purely explicit methods may not be feasible. Both SDFs and NERFs find applications in various robotics tasks. We present two examples below :
>
> - **Truncated Signed Distance Fields** were recently successfully applied to create 3D-fields for Robotic manipulation (re-arrangement problem) (applying large VLMs in zero-shot fashion + MPC on learned dynamics) [1] .
> - In [2], the authors used a **NERF** to create reconstructions of an office environment.
>
> Thus our main focus in this work is on INRs. Our MAG layer is designed to reduce both inference time and memory requirements with only a minimal drop in quality and thus replacing MLPs by our layers can speed up INRs. For models with a large number of MLPs, like NERF, iSDF, ViT and MIP-NERF, we consistently get at least $20$% gains inference speeds with minimal drop in quality.
>
> To summarize : For NeRF, we achieve a **24%** improvement. Similarly, iSDF shows a **23%** gain. Moreover, for ViT, we achieve nearly a **35%** boost in inference speed. For MIP-NERF, we show an improvement of **18%**. Overall, the quality drop across all cases is consistently minimal (about 2%).
>
> Furthermore these gains for NERF are higher than some existing published work (for ex: see [3] where the inference gain is around 18% with a quality drop of 3%).
>
> [1] $D^3$ Fields: Dynamic 3D Descriptor Fields for Zero-Shot Generalizable Rearrangement. Wang et al. CoRL 2024
>
> [2] Mobility VLA: Multimodal Instruction Navigation with Long-Context VLMs and Topological Graphs. Chiang et al.  CoRL 2024
>
> [3] CoordX: Accelerating Implicit Neural Representation with a Split MLP Architecture. Liang et al. ICLR 2022
>
> > *A simple baseline by reducing the number of layers or matching the training parameters*
>
> Thank you for proposing this experiment. We added these results in Fig 5. We show that our models offer similar or higher reconstruction quality than the models with the same number of training parameters while being consistently faster. The speed-ups are achieved due to the compression trick as explained in Appendix which allows us to combine any subsequent FFL. This results in effectively converting a 2-layer neural network into a single layer.  For NERF, we observe that reducing the size of the model to match our parameter count results in **2 %** degradation in quality, while still being almost **5 %** slower.
>
> Furthermore, we also show that reducing the layer size in iSDF to match the number of training parameters in our proposed MAG-iSDF results in a **4.5%** performance decrease compared to MAG-iSDF. The smaller iSDF is also **2%** slower than our MAG-iSDF due to our novel compression trick. (see Fig 7).
>
> Thus simple speed-ups techniques do not work quite as well as our MAG-layers which can retain quality as well as achieve speedups.
>
>
> > *Other tasks like classification using Transformers.*
>
> Thank you for proposing this experiment. In this setting, we replace a part of the Feed-Forward Network (FFN) block in Vision Transformer (ViT) with the MAG layer, namely the expansion layer with the GeLU activation. Note that : the MAG layer can then be combined with the following linear layer to create a single linear layer of size $[r, 768]$, where $r$ is the number of random features. This allows for a significant compression of the ViT model. To summarize our findings : by replacing top-6 layer’s MLP blocks with MAG reduces the size of the ViT model from 346 Mb to 196.85 Mb. This speeds up inference by almost $\mathbf{35}$% and has minimal impact on accuracy (less than 2%). We present more details in Appendix E Image Classification Experiments. For convenience, we also show the results comparing inference parameters (in millions) vs accuracy in the table below :
>
> | Dataset      | 86 (Full Fine-tuning)    | 82     | 77     | 72     | 67     | 62     | 57     |
> |--------------|--------|--------|--------|--------|--------|--------|--------|
> | CIFAR-10     | 98.95  | 98.92  | 98.84  | 98.76  | 98.61  | 98.14  | 97.56  |
> | CIFAR-100    | 91.67  | 91.65  | 91.24  | 90.87  | 90.13  | 89.71  | 89.45  |
> | DTD    | 75.0  | 76.23  | 75.61  | 75.12  | 74.86  | 74.31  | 73.94  |

---

> > ### Author Response · Authors · 2024-11-19
> >
> > > *Improved Results*
> >
> > Recent works [1,2] have shown improved performance when the projection matrix $\mathbf{G}$ can be trained instead of sampled from a fixed distribution. We show that it is indeed true in our case as well. Note that this incurs no additional cost in inference. In all cases, including NERF, D-NERF, MipNERF, ZipNERF, iSDF, ViT we show that using trainable $\mathbf{G}$, our performance drop is minimal and we have updated all the tables in the paper with these new results. For convenience, we summarize them in the table below :
> >
> > | Model      | Delta in Performance (%) | Delta in Inference Speed (%) |
> > |------------|---------------------------|-------------------------------|
> > | NERF       | -1                         | +24                            |
> > | iSDF       | -4                         | +23                            |
> > | D-NERF     | - <1                        | +6                             |
> > | Zip-NERF   | -1                         | +7                             |
> > | ViT        | -2                         | +35                            |
> > | Mip-NERF   |   -2                        |           + 18                    |
> >
> > Thus our method becomes quite powerful in INRs and in models where there are a lot of MLP layers.
> >
> > One of the core contributions of our method is that it can be swapped directly inside a pre-trained INR during inference via a simple layer wise distillation. Our distillation method is actually a real world test case of Q1 where the distributions are coming from pretrained models. Our MAG-layers can approximate complex outputs well enough that they can simply replace FFLs, thus justifying the design choice of $f$. We show that using a trainable G while using distillation, one can recover the performance of various INRs. Thus this simple layer wise distillation method is supposed to be a test-time inference speed up method where one does not have access to a lot of resources. This distillation takes less than a second and very limited GPU resources to train. Our new results on layer-wise distillation can actually maintain quality while reducing inference speeds (see Fig 8 and 9).
> >
> > [1] On Learning the Transformer Kernel. Chowdhury et al. 2022
> >
> > [2] The Hedgehog & the Porcupine: Expressive Linear Attentions with Softmax Mimicry. Zhang et al. ICLR 2024
> >
> >
> > > *Trade-off than a true improvement*
> >
> > We appreciate the opportunity to address the trade-offs involved in our work. These primarily arise from the number of random features and the number of feed-forward layers (FFL) replaced by our MAG-layers. However, we believe these trade-offs represent meaningful advancements. Notably, our MAG-based algorithms remain the fastest during inference compared to alternative strategies, such as applying more shallow models (as demonstrated in our additional rebuttal experiments). Moreover, we would like to emphasize that trade-offs often underpin significant progress. For instance, the extensive literature on efficient-attention mechanisms for Transformers [1, 2, 3, 4, 5] is built on carefully balancing quality and speed through various strategies, including low-rank and sparse attention, as well as hashing and clustering methods.
> >
> >
> > Finally, we would like to point out the following novelties of our work :
> >
> > - Thinking of FFL computation as a kernel method is a new method and it comes with theoretical guarantees with principled variance reduction techniques.
> > - Reimagining the computation as a kernel method allows us to combine a MAG linear with **any** subsequent FFL, effectively compressing a 2-layer network into a single layer.
> > - This new method is completely orthogonal to all techniques that are used to commonly speed computation of FFLs or INRs in particular, and so can be combined with them.
> > - This layer is general in nature and can actually be used to reduce the computation burden of models where there are a lot of FFLs.
> > - The number of random features is a hyperparameter and can be used to trade-off quality vs time or memory.
> > - **Expressiveness:** Our MAG mechanism is not contained within the regular FFL framework, and is capable of modeling relationships beyond those of standard FFLs.
> >
> > [1] Rethinking Attention with Performers. Choromanski et al. ICLR 2021
> >
> > [2] Linformer: Self-Attention with Linear Complexity. Wang et al. 2020
> >
> > [3] Longformer: The long-document transformer. Beltagy et al. 2020.
> >
> > [4] Generating long sequences with sparse transformers. Child et al. 2019
> >
> > [5] Simple linear attention language models balance the recall-throughput tradeoff. Arora et al. 2024

---

> ### Comment · Reviewer_TtzD · 2024-11-21
>
> I appreciate the authors' detailed response to my comments. However, I still have a few remaining concerns.
>
> The claim that "Our method is 1% behind the baseline in terms of PSNR but 24% faster in inference speed" is problematic. PSNR is defined in the logarithmic domain, making it inappropriate to directly compare percentage differences between PSNR and inference speed. Moreover, the newly provided Figure 5 indicates that, when the number of parameters is the same, the performance of the original MLP and the proposed MAG layer is very similar, with differences so minor that they are unlikely to have any meaningful impact.
>
> As a result, I find the significance of this work within the context of 3D reconstruction, which is the primary focus of this paper as suggested by its title, to be quite limited. That said, this work might have greater relevance in the area of neural network design. However, to make such a claim, the authors would need to rethink the narrative of their paper and include comparisons to other neural networks such as KAN. This is outside my area of expertise, but I personally believe this work might be better positioned in the field of neural network design, where 3D reconstruction serves as just one application of the proposed method.

---

> > ### Author Response · Authors · 2024-11-23
> >
> > Dear Reviewer,
> >
> > We sincerely thank you for your feedback and comments.
> >
> >  Best,
> >
> >  The Authors.

---

### Author Response · Authors · 2024-11-19
**Summary of the changes**

We thank the reviewers for taking the time to go through our paper and providing detailed feedback/suggestions. We have conducted additional experiments and revised the paper to address any concerns. Here is the summary of all the changes:


- We have added new image classification experiments using ViT. (Appendix E)
- We have improved the results on **all** our experiments by allowing the projections to be trainable instead of sampled from a fixed distribution and have correspondingly updated all the tables with the results.
- In each experiment, we made clear what the trade-offs are : efficiency vs quality.
- We have provided ablations for each experiment by reducing the parameter count of the base models to match our MAG-models.
- We added a new INR experiment on MIP-NERF.

For convenience, we highlighted the main changes in blue in our manuscript. Please let us know if you have concerns or questions.

---

### Meta-Review · Area_Chair_CxfJ · 2024-12-19

**Metareview:**

This submission proposes a new network layer, towards enabling efficiency improvements for implicit neural representation (INR) tasks. The paper draws on inspiration from kernel method approximations, where input magnitudes are considered in order to disentangle them from MLP layer weights. Experimental work evidences improvements in inference speed and model size, across various tasks.

The paper received reviews from three reviewers that collectively note core idea technical-soundness and thorough experimental investigation. The crux of the shared common concerns relate to the limited significance of the work, within the context of 3D reconstruction, and the potential low impact. Multiple reviewers suggest to generalise and better position the contributions, as applicable beyond INR.

AC believes that additions and narrative refinements resuting from the rebuttal can likely benefit from a further round of review and therefore recommends rejection. Authors are encouraged to use the collective feedback to strengthen the work towards submission at an alternative venue.

**Additional Comments On Reviewer Discussion:**

The paper received reviews from three reviewers that collectively note core idea technical-soundness and thorough experimental investigation. The crux of the shared common concerns relate to the limited significance of the work, within the context of 3D reconstruction, and the potential low impact. Multiple reviewers suggest to generalise and better position the contributions, as applicable beyond INR.

The rebuttal alleviates technical queries concerning kernel-method and dictionary-learning connections, allowing one reviewer to lean towards acceptance yet stops short of championing the paper. The majority of reviewers remain unconvinced and somewhat luke-warm regarding the author response, opting to retain negative views that list significance, scoping + impact concerns. Minor technical discussion points also remain open.

Authors further respond with explorations of (i) application to models with larger MLP counts and (ii) trainable projections (c.f. fixed distribution sampling) which are interesting yet may require further refinement.

---

### Decision · Program_Chairs · 2025-01-22

Reject